# Bounded Space Differentially Private Quantiles

**Daniel Alabi**                                             *alabid@cs.columbia.edu*
*Columbia University*

**Omri Ben-Eliezer**                                              *omrib@mit.edu*
*Massachusetts Institute of Technology*

**Anamay Chaturvedi**                                  *chaturvedi.a@northeastern.edu*
*Northeastern University*

**Reviewed on OpenReview:** *https://openreview.net/forum?id=sixOD8YVuM*

## Abstract

Estimating the quantiles of a large dataset is a fundamental problem in both the streaming algorithms literature and the differential privacy literature. However, all existing private mechanisms for distribution-independent quantile computation require space at least linear in the input size $n$. In this work, we devise a differentially private algorithm for the quantile estimation problem, with strongly sublinear space complexity, in the one-shot and continual observation settings. Our basic mechanism estimates any $\alpha$-approximate quantile of a length-$n$ stream over a data universe $\mathcal{X}$ with probability $1 - \beta$ using $O\left(\frac{\log(|\mathcal{X}|/\beta)\log n}{\alpha\epsilon}\right)$ space while satisfying $\epsilon$-differential privacy at a single time point. Our approach builds upon deterministic streaming algorithms for non-private quantile estimation instantiating the exponential mechanism using a utility function defined on sketch items, while (privately) sampling from intervals defined by the sketch. We also present another algorithm based on histograms that is especially well-suited to the multiple quantiles case. We implement our algorithms and experimentally evaluate them on synthetic and real-world datasets.

## 1 Introduction

Quantile estimation is a fundamental subroutine in data analysis and statistics. For $q \in [0, 1]$, the $q$-quantile of a dataset of size $n$ is the element ranked $\lceil qn \rceil$ when the elements are sorted from smallest to largest. Computing a small number of quantiles in a massive data set can serve as a quick and effective sketch of the "shape" of the data. Quantile estimation also serves an essential role in robust statistics, where data is generated from some distribution but is contaminated by a non-negligible fraction of outliers, i.e., "out of distribution" elements that may sometimes even be adversarial. For example, the median (50th percentile) of a dataset is used as a robust estimator of the mean in such situations where the data may be contaminated. Location parameters can also be (robustly) estimated via truncation or winsorization, an operation that relies on quantile estimation as a subroutine (Tukey, 1960; Huber, 1964). Rank-based nonparametric statistics can be used for hypothesis testing (e.g., the Kruskal-Wallis test statistic (Kruskal & Wallis, 1952)). Thus, designing quantile-based or rank-based estimators, whether distribution-dependent or distribution-agnostic, is important in many scenarios.

Maintaining the privacy of individual users or data items, or even of groups, is an essential prerequisite in many modern data analysis and management systems. Differential privacy (DP) is a rigorous and now well-accepted definition of privacy for data analysis and machine learning. In particular, there is already a substantial amount of literature on differentially private quantile estimation (e.g., see (Nissim et al., 2007; Asi & Duchi, 2020; Gillenwater et al., 2021; Tzamos et al., 2020)).[1]

---

[1]Robust estimators are also known to be useful for accurate differentially private estimation; see, e.g., the work of Dwork and Lei (Dwork & Lei, 2009) in the context of quantile estimation for the interquartile range and for medians.

All previous work either makes certain distributional assumptions about the input, or assumes the ability to access all input elements (thus virtually requiring a linear or worse space complexity). Such assumptions may be infeasible in many practical scenarios, where large scale databases have to quickly process streams of millions or billions of data elements without clear a priori distributional characteristics. The field of streaming algorithms aims to provide space-efficient algorithms for data analysis tasks such as these. These algorithms typically maintain good accuracy and fast running time while having space requirements that are substantially smaller than the size of the data. While distribution-agnostic quantile estimation is among the most fundamental problems in the streaming literature (Agarwal et al., 2013; Felber & Ostrovsky, 2017; Greenwald & Khanna, 2001; Hung & Ting, 2010; Karnin et al., 2016; Manku et al., 1999; Munro & Paterson, 1980; Shrivastava et al., 2004; Wang et al., 2013), no differentially private sublinear-space algorithms for the same task are currently known. Thus, the following question, essentially posed by (Smith, 2011) and (Mir et al., 2011), naturally arises:

> *Can we design differentially private quantile estimators that use space sublinear in the stream length, have efficient running time, provide high-enough utility, and do not rely on restrictive distributional assumptions?*

It is well-known (Munro & Paterson, 1980) that exact computation of quantiles cannot be done with sublinear space, even where there are no privacy considerations. Thus, one must resort to approximation. Specifically, for a dataset of $n$ elements, an $\alpha$-*approximate q-quantile* is any element which has rank $(q \pm \alpha)n$ when sorting the elements from smallest to largest, and it is known that the space complexity of $\alpha$-approximating quantiles is $\tilde{\Omega}(1/\alpha)$ (Munro & Paterson, 1980). In our case, the general goal is to efficiently compute $\alpha$-approximate quantiles in a (pure or approximate) differentially private manner.

## 1.1 Our Contributions

We answer the above question affirmatively by providing theoretically proven algorithms with accompanying experimental validation for quantile estimation with DP guarantees. The algorithms are suitable for private computation of either a single quantile or multiple quantiles. Concretely, the main contributions are:

1. We devise `DPExpGK`, a differentially private sublinear-space algorithm for quantile estimation based on the exponential mechanism. In order to achieve sublinear space complexity, our algorithm carefully instantiates the exponential mechanism with the basic blocks being intervals from the Greenwald-Khanna (Greenwald & Khanna, 2001) data structure for non-private quantile estimation, rather than single elements. We prove general distribution-agnostic utility bounds on our algorithm and show that the space complexity is logarithmic in $n$.

2. We present `DPHistGK`, another differentially private mechanism for quantile estimation, which applies the Laplace mechanism to a histogram, again using intervals of the GK-sketch as the basic building block. We theoretically demonstrate that `DPHistGK` may be useful in cases where one has prior knowledge on the input.

3. We extend our results to the *continual release* setting, wherein we must maintain and output an estimate of the queried quantile in an online manner as the data is received and processed.

4. We empirically validate our results by evaluating `DPExpGK`, analyzing and comparing various aspects of performance on real-world and synthetic datasets.

## 2 Related Work

### 2.1 Quantile Approximation of Streams and Sketches

Approximation of quantiles in large data streams (without privacy guarantees) is among the most well-investigated problems in the streaming literature (Wang et al., 2013; Greenwald & Khanna, 2016; Xiang et al., 2020). A classical result of Munro and Paterson from 1980 (Munro & Paterson, 1980) shows that

computing the median exactly with $p$ passes over a stream requires $\Omega(n^{1/p})$ space, thus implying the need for approximation to obtain very efficient (that is, at most polylogarithmic) space complexity. Manku, Rajagopalan and Lindsay (Manku et al., 1999) built on ideas from (Munro & Paterson, 1980) to obtain a randomized algorithm with only $O((1/\alpha)\log^2(n\alpha))$ for $\alpha$-approximating all quantiles; a deterministic variant of their approach with the same space complexity exists as well (Agarwal et al., 2013). The best known deterministic algorithm is that of Greenwald and Khanna (GK) (Greenwald & Khanna, 2001) on which we build on in this paper, with a space complexity of $O(\alpha^{-1}\log(\alpha n))$ to sketch all quantiles for $n$ elements (up to rank approximation error of $\pm\alpha n$). A recent deterministic lower bound of Cormode and Veselý (Cormode & Veselý, 2020) (improving on (Hung & Ting, 2010)) shows that the GK algorithm is in fact optimal among deterministic (comparison-based) sketches.

Randomization and sampling help for streaming quantiles, and the space complexity becomes independent of $n$; an optimal space complexity of $O((1/\alpha)\log\log(1/\beta))$ was achieved by Karnin, Lang and Liberty (Karnin et al., 2016) (for failure probability $\beta$), concluding a series of work on randomized algorithms (Agarwal et al., 2013; Felber & Ostrovsky, 2017; Luo et al., 2016; Manku et al., 1999).

The problem of biased or relative error quantiles, where one is interested in increased approximation accuracy for very small or very large quantiles, has also been investigated (Cormode et al.; 2006); it would be interesting to devise efficient differentially private algorithms for this problem.

Recall that our approach is based on the Greenwald-Khanna deterministic all-quantiles sketch (Greenwald & Khanna, 2001). While some of the aforementioned randomized algorithms have a slightly better space complexity, differential privacy mechanisms are inherently randomized by themselves, and the analysis seems somewhat simpler and more intuitive when combined with a deterministic sketch. This, of course, does not rule out improved private algorithms based on modern efficient randomized sketches (see Section 7).

## 2.2 Differential Privacy

**Differentially private single quantile estimation:** Works by Nissim, Raskhodnikova and Smith (Nissim et al., 2007) and Asi and Duchi (Asi & Duchi, 2020) improve the trade-off between accuracy and privacy by scaling the noise added for obfuscation in an instance-specific manner for median estimation in the absence of distributional assumptions. Another work by Dwork and Lei (Dwork & Lei, 2009) uses a "propose-test-release" paradigm to take advantage of local sensitivity; however, as observed in Gillenwater et al. (2021), in practice the error incurred by this method is relatively large as compared to other works like Tzamos et al. (2020). The work of Tzamos et al. (2020) achieves the optimal trade-off between privacy and utility in the distributional setting, but again as observed by Gillenwater et al. (2021), with a time complexity of $O(n^4)$, this method does not scale well to large data sets.

The work of Gillenwater et al. (2021) shows how to optimize the division of the privacy budget to estimate $m$ quantiles in a time-efficient manner. For estimation of $m$ quantiles, their time and space complexity are $O(mn\log(n) + m^2n)$ and $O(m^2n)$, respectively. They do an extensive experimental analysis and find lower error compared to previous work. However, although they provide intuition for why their method should incur relatively low error, they do not achieve formal theoretical accuracy guarantees. Kaplan et al. (2022) improve upon the work of Gillenwater et al. (2021) for the multiple approximate quantiles problem by using a tree-based (recursive) approach to CDF estimation. However, none of these works deal with quantile estimation in the sublinear space setting. Böhler & Kerschbaum (2020) solve the problem of estimating the joint median of two private data sets with time complexity sub-linear in the size of the data-universe and provide privacy guarantees for small data sets as well as limited group privacy guarantees unconditionally against polynomially time-bounded adversaries.

**Inherent privacy:** Another line of work (Blocki et al., 2012; Smith et al., 2020; Choi et al., 2020) demonstrates that sketching algorithms for streaming problems might have inherent privacy guarantees under minimal assumptions on the dataset in some cases. For such algorithms, relatively little noise needs to be added to preserve privacy unconditionally.

# 3 Preliminaries and Notation

We now give standard differential privacy notation and formally describe the quantile estimation problem. We also present the Greenwald-Khanna sketch guarantees in a suitable form.

## 3.1 Differential Privacy

**Definition 3.1** (Differential Privacy (Dwork et al., 2006)). Let $\mathcal{Q} : \mathcal{X}^n \to \mathcal{R}$ be a (randomized) mechanism. For any $\epsilon \geq 0, \delta \in [0, 1]$, $\mathcal{Q}$ satisfies $(\epsilon, \delta)$**-differential privacy** if for any neighboring databases (that differ in one row) $\mathbf{x} \sim \mathbf{x}' \in \mathcal{X}^n$ and any set $S \subseteq \mathcal{R}$,

$$\mathbb{P}[\mathcal{Q}(\mathbf{x}) \in S] \leq e^\epsilon \mathbb{P}[\mathcal{Q}(\mathbf{x}') \in S] + \delta.$$

The probability is taken over the coin tosses of $\mathcal{Q}$. We say that $\mathcal{Q}$ satisfies pure differential privacy ($\epsilon$-DP) if $\delta = 0$ and approximate differential privacy (($\epsilon, \delta$)-DP) if $\delta > 0$. We can set $\epsilon$ to be a small constant (e.g., between 0.01 and 2) but will require that $\delta \leq n^{-\omega(1)}$ be cryptographically small.

## 3.2 Quantile Approximation

In this subsection we make some definitions to formalize our analysis.

**Definition 3.2.** (Quantiles) Let there be a totally ordered data universe $\mathcal{X}$ and an input data stream $X = ((x_1, 1), \ldots, (x_n, n))$ (sometimes implicitly referred to as $X = (x_1, \ldots, x_n)$). For $(x_i, i) \in X$, let $\mathrm{val}((x_i, i)) = x_i$ and $\mathrm{ix}((x_i, i)) = \sum_{j \leq i} |\{(x_j, j) : x_j < x_i \text{ or } x_j = x_i, j < i\}|$. We abuse notation to say that for $v_1, v_2 \in X$, $v_1 \leq v_2$ if $\mathrm{ix}(v_1) \leq \mathrm{ix}(v_2)$. Then, the $q$-quantile of $X$ is $\mathrm{val}(v)$ for $v \in X$ with $\mathrm{ix}(v) = \lceil qn \rceil$.

We observe that in this setting a value from the data universe can occur multiple times in a set. To account for this we define a range of ranks that any value can hold; this will be useful when reasoning about quantile approximation.

**Definition 3.3.** (Rank and approximate quantiles) For $x \in \mathcal{X}$, we define $r_{\min}(x) = |\{v \in X : \mathrm{val}(v) < x\}|$, $r_{\max}(x) = |\{v \in X : \mathrm{val}(v) \leq x\}|$ and $\mathrm{rank}(X, x)$ to be the interval $[r_{\min}(x), r_{\max}(x)]$. We say that $x \in \mathcal{X}$ is an $\alpha$-approximate $q$-quantile for $X$ if $\mathrm{rank}(X, x) \cap [\lceil qn \rceil - \alpha n, \lceil qn \rceil + \alpha n] \neq \emptyset$.

Figure 1: In the data set $S'$ of 8 elements, the true 0.5 quantile is the value 2, and the values $2, 3, 4$ and 5 are all acceptable 0.25-approximate answers. Note that although 1 is adjacent to 2 in the data universe $\mathcal{X}$, it is not an acceptable output.

Consider the following example to see how these definitions play out in practice.

**Example 3.4.** Given the data set $\{1, 2, 2, 3, 5, 2, 6, 5\}$ (refer to figure 1), the 0.5 quantile is 2, which we distinguish from the median, which would be the average of the elements ranked 4 and 5, i.e, 2.5 for this data set. A straightforward way to obtain quantiles is to sort the dataset and the pick the element at the $\lceil q \cdot n \rceil$ position. This method only works in the offline (non-streaming) setting. For $\alpha = 0.25$, the $\alpha$-approximate 0.5 quantiles are $2, 3, 4$ and 5. Note that 4, which does not occur in the data set, is still a valid response, but 1, which occurs in the data set and is even adjacent to the true 0.5 quantile 2 in the data universe $\mathcal{X}$, is not a valid response.

We can now formalize the two versions of the problems that are dealt with in the literature on streaming quantile approximation. Let $\mathcal{D}$ be any distribution with random variable $X \sim \mathcal{D}$ and CDF $F_X : \mathbb{R} \to [0, 1]$. For any $q \in [0, 1]$, $Q_{\mathcal{D}}^q$ denotes the value $x$ such that $F_X(x) = q$.

**Definition 3.5** (Single Quantile). Given sample $S = (x_1, \ldots, x_n)$ in a streaming fashion in arbitrary order, construct a data structure for computing the quantile $Q_{\mathcal{D}}^q$ such that for any $q \in [0, 1]$, with probability at least $1 - \beta$, $|Q_{\mathcal{D}}^q - \tilde{Q}_S^q| \leq \alpha$.

**Definition 3.6** (All Quantiles). Given sample $S = (x_1, \ldots, x_n)$ in a streaming fashion in arbitrary order, construct a data structure for computing the quantile $Q_{\mathcal{D}}^q$ such that with probability at least $1 - \beta$, for all values of $q \in M$ where $M \subset [0, 1]$, $|Q_{\mathcal{D}}^q - \tilde{Q}_S^q| \leq \alpha$.

### 3.3 Non-private Quantile Streaming

In this subsection, we provide formal guarantees needed for any sketch that our algorithms can build upon:

**Lemma 3.7.** *Given a data stream $x_1, \ldots, x_n$ of elements drawn from $\mathcal{X}$, there exists a sketching algorithm that outputs a list $S(X)$ of $s = O\left(\frac{1}{\alpha} \log \alpha n\right)$ many tuples $(v_i, g_i, \Delta_i) \in (\mathcal{X} \times \mathbb{N}) \times \mathbb{N} \times \mathbb{N}$ for $i = 1, \ldots, s$ such that if $X$ is the data multi-set then:*

1. *$\mathrm{rank}(X, \mathrm{val}(v_i)) \subset [\sum_{j \leq i} g_j, \Delta_i + \sum_{j \leq i} g_j]$.*

2. *$g_i + \Delta_i \leq 2\alpha n$.*

3. *The first tuple is $(\min\{x \in X\}, 1, 0)$ and the last tuple is $(\max\{x \in X\}, 1, 0)$.*

4. *The $v_i$ are sorted in ascending order. Without loss of generality, the lower interval bounds $\sum_{j \leq i} g_j$ and upper interval bounds $\Delta_i + \sum_{j \leq i} g_j$ are also sorted in increasing order.*

From conditions 1 and 2 above, it follows that the GK sketch can be used to compute $\alpha$-approximate $q$-quantiles by outputting the value $v_{i^*}$ for the unique index $i^*$ such that $\sum_{j \leq i^*} g_j \leq qn \leq \Delta_{i^*} + \sum_{j \leq i^*} g_j$. More generally, any sketch which achieves the properties outlined in Lemma 3.7 can be used in place of GK.

## 4 Differentially Private Algorithms

In this section we present our two DP mechanisms for quantile estimation. Throughout, we assume that $\alpha$ is a user-defined approximation parameter. The goal is to obtain $\alpha$-approximate $q$-quantiles.

The new algorithms we introduce are:

(1) `DPExpGK` (Algorithm 1): An exponential mechanism based $(\epsilon, 0)$-DP algorithm for computing a single $q$-quantile. To solve the all-quantiles problem with approximation factor $\alpha$, one can run this algorithm iteratively with target quantile $0, \alpha, 2\alpha, \ldots$. Doing so requires scaling the privacy parameter in each call by an additional $\alpha$ factor which increases the space complexity by a factor of $1/\alpha$. We also give an optimized implementation (algorithm 8) of this algorithm in the appendix. We extend our results to the continual observation setting (Dwork et al., 2010; Chan et al., 2011).

(2) `DPHistGK` (Algorithm 2): A histogram based $(\epsilon, 0)$-DP algorithm (Algorithm 2) for the $\alpha$-approximate all-quantiles problem. The privacy guarantee of this algorithm is unconditional, but there is no universal theoretical utility bound as in the previous algorithm. However, in some cases the utility is provably better: for example, we show that if the data set is drawn from a normal distribution (with unknown mean and variance), we can avoid the quadratic $1/\alpha^2$ factor in the sample complexity that we incur when using `DPExpGKGumb` for the same all-quantiles task.

The rest of this section contains the descriptions of the two algorithms; we relegate all proofs to the appendix.

### 4.1 `DPExpGKGumb`: Exponential Mechanism Based Approach

We first establish how the exponential mechanism and the GK sketch may be used in conjunction to solve the single quantile problem. Concretely, the high level idea is to call the privacy preserving exponential mechanism with a score function derived from the GK sketch. The exponential mechanism is a fundamental privacy primitive which when given a public set of choices and a private score for each choice outputs a choice

---

**Algorithm 1:** `DPExpGK`: Exponential Mechanism DP Quantiles : High Level Description

---

**Data:** $X = (x_1, x_2, \ldots, x_n)$

**Input:** $\epsilon, \alpha$ (approximation parameter), $q \in [0, 1]$ (quantile parameters), $\delta_u$ (sensitivity)

**1** $S(X) = \{(v_i, g_i, \Delta_i) : i \in [s]\} \leftarrow GK(X, \alpha)$

**2** Define score function

$$u(S(X), x) = -\min\{|y - \lceil qn \rceil| : y \in [\hat{r}_{\min}(x), \hat{r}_{\max}(x)]\}. \tag{1}$$

where

$$\hat{r}_{\min}(x) = \max\{\sum_{j \le i} g_j : \text{val}(v_i) < x\}$$

$$\hat{r}_{\max}(x) = \min\{\Delta_i + \sum_{j \le i} g_j : \text{val}(v_i) > x\}$$

**3** Execute the exponential mechanism with score function $u(S(X), \cdot)$, i.e. choose and output a single $e \in \mathcal{X}$ with probability

$$\propto \exp\left(\frac{\epsilon}{2\delta_u} \cdot u(S(X), e)\right).$$

---

that with high probability has a score close to optimal whilst preserving privacy. In the course of constructing our algorithms, we have to resolve two problems; one, how to usefully construct a score function to pass to the exponential mechanism so that the private value derived is a good approximation to the $q$-quantile, and two, how to execute the exponential mechanism efficiently on the (possibly massive) data universe $\mathcal{X}$. To resolve the first issue we devise the (not necessarily efficient) routine Algorithm 1; and to resolve the second one, we run an essentially equivalent but far more efficient routine, Algorithm 8.

**Constructing a score function:** We recall that the GK sketch returns a short sequence of elements from the data set with a deterministic interval for their ranks and the promise that for any target quantile $q \in [0, 1]$, there is some sketch element that lies within $\alpha n$ units in rank of $\lceil qn \rceil$. One technicality that we run into when trying to construct a score function on the data universe $\mathcal{X}$ is that when a single value occurs with very high frequency in the data set, the ranks of the set of occurrences can span a large interval in $[0, n]$, and there is no one rank we can ascribe to it so as to compare it with the target rank $\lceil qn \rceil$. This can be resolved by defining the score for any data domain value in terms of the distance of its respective rank interval $[r_{\min}(x), r_{\max}(x)]$ (formalized in Definition B.1) from $\lceil qn \rceil$; elements whose intervals lie closer to the target have a higher score than further away ones.

**Efficiently executing the exponential mechanism:** The exponential mechanism samples one of the public choices (in our case some element from the data universe $\mathcal{X}$) with probability that increases with the quality of the choice according to the score function. In general every element can have a possibly different score and the efficiency of the exponential mechanism can vary widely depending on the context. In our setting, the succinctness of the GK sketch leads to a crucial observation: by defining the score function via the sketch, the data domain is partitioned into a relatively small number of sets such that the score function is constant on each partition. Concretely, for any two successive elements in the GK sketch, the range of values in the data universe that lie between them will have the same score according to our score function. We can hence first sample a partition from which to output a value, and then choose a value from within that interval uniformly at random. To make our implementation even more efficient and easy to use, we also make use of the *Gumbel-max* trick that allows us to iterate through the set of choices instead of storing them in memory - see appendix B.1 and the expanded pseudocode in algorithm 8 for more detail.

On formalizing this outline we get the following formal guarantees for algorithm 1 (please see appendix B.1 for a complete proof). $d(\cdot, \cdot)$ denotes the $\ell_1$ metric on $\mathbb{R}$.

**Theorem 4.1.** *Algorithm 1 is $\epsilon$-differentially private. Let $\hat{x}$ be the value returned by Algorithm 1 when initialized with target quantile q. The following statements hold:*

1. *Algorithm 1 can be implemented to run with space complexity $O((1/\alpha)\log \alpha n)$, such that with probability $1 - \beta$*

$$d(\lceil qn \rceil, [\hat{r}_{\min}(\hat{x}), \hat{r}_{\max}(\hat{x})]) \leq 2\alpha n + \frac{2(4\alpha n + 2)\log(|\mathcal{X}|/\beta)}{\epsilon} \tag{2}$$

2. *For $n > \frac{24\log|\mathcal{X}|/\beta}{\alpha \min\{\epsilon,1\}}$, Algorithm 1 can be implemented to run with space complexity $O\left((\alpha\epsilon)^{-1}\log(|\mathcal{X}|/\beta)\log n\right)$ such that with probability $1 - \beta$, $\hat{x}$ is an $\alpha$ approximate q-quantile.*

### 4.2 `DPHistGK`: Histogram Based Approach

For methods in this section, we assume that we have $K \geq 1$ *disjoint* bins each of width $w$ (e.g., $w = \alpha/2$) partitioning the data universe. These bins are used to construct a histogram. Essentially, Algorithm 2 builds an empirical histogram based on the GK sketch, adds noise so that the bin values satisfy $(\epsilon, 0)$-DP, and converts this empirical histogram to an approximate empirical CDF, from which the quantiles can be approximately calculated. We demonstrate one use case of `DPHistGK` where the space complexity required improves upon the worst-case bound for `DPExpGK`, Theorem 4.1. While the histogram based mechanism does not have universal utility bounds in the spirit of the above theorem, the results in this section serve as one simple example where it may yield desirable accuracy while using less space.

---

**Algorithm 2:** `DPHistGK`: Computing DP Quantiles in Bounded Space

**Data:** $X = (x_1, x_2, \ldots, x_n)$
**Input:** $\epsilon, \alpha$ (approximation parameter), $q \in [0, 1]$ (quantile parameters), $w$
**1** Build summary sketch $S(X)$ where
**2** $S(X) = \{(v_i, g_i, \Delta_i) : i \in [s]\} \leftarrow GK(X, \alpha)$
  /* cell labels $a_i$ and counts $c_i = 0$                                                          */
**3** Initialize data-agnostic (empty) histogram $Hist = \langle (a_i, c_i), \ldots \rangle$ with cell widths $w$
**4 for** $(v_i, g_i, \Delta_i) \in S(X)$ **do**
**5** | Insert $g_i$ counts of $v_i$ into histogram $Hist$

**6** $c = 0$
**7** $H = [\cdot]$/* initialize empty list                                                          */
**8 for** $(a_i, c_i) \in Hist$ **do**
**9** | $\tilde{c}_i = \max(0, c_i + \text{Lap}(0, 2/\epsilon))$
**10** | Append $(a_i, c + \tilde{c}_i)$ to $H$
**11** | $c = c + \tilde{c}_i$

**12** $r = \lceil q \cdot n \rceil$
**13 for** $(b, rank) \in H$ **do**
**14** | **if** $r < rank$ **then**
**15** | | **return** $b$

  /* return last element of $H$                                                                      */
**16 return** $H[|H| - 1]$

---

Suppose that we are given an i.i.d. sample $S = (X_1, X_2, \ldots, X_n)$ such that for all $i \in [n]$, $X_i \sim \mathcal{N}(\mu, \sigma^2 I_{d \times d})$, $\mu \in \mathbb{R}^d$. The goal is to estimate DP quantiles of the distribution $\mathcal{N}(\mu, \sigma^2 I_{d \times d})$ without knowledge of $\mu$ or $\sigma^2$. We will show how to estimate the quantiles assuming that $\sigma^2$ is known. Note that it is easy to generalize the work to the case where $\sigma^2$ is unknown as follows:

For any sample $S$ drawn from i.i.d. from $\mathcal{N}(\mu, \sigma^2 I_{d \times d})$, the $1 - \beta$ confidence interval is

$$\bar{X} \pm \frac{\sigma}{\sqrt{n}} \cdot z_{1-\beta/2},$$

where $z_{1-\beta/2}$ is the $1 - \beta/2$ quantile of the standard normal distribution and $\bar{X}$ is the empirical mean. The length of this interval is fixed and equal to $\frac{2\sigma z_{1-\beta/2}}{\sqrt{n}} = \Theta\left(\frac{\sigma}{\sqrt{n}}\sqrt{\log\frac{1}{\beta}}\right)$. In the case where $\sigma^2$ is unknown, the confidence interval becomes $\bar{X} \pm \frac{s}{\sqrt{n}} \cdot t_{n-1,1-\beta/2}$, where $s^2 = \frac{1}{n-1}\sum_{i=1}^{n}(X_i - \bar{X})^2$ is the sample variance (sample estimate of $\sigma^2$) and $t_{n-1,1-\beta/2}$ is the $1 - \beta/2$ quantile of the $t$-distribution with $n - 1$ degrees of freedom. The length of the interval can be shown to be

$$\frac{2\sigma}{\sqrt{n}} \cdot k_n \cdot t_{n-1,1-\beta/2} = \Theta\left(\frac{\sigma}{\sqrt{n}}\sqrt{\log\frac{1}{\beta}}\right),$$

where $k_n = 1 - O(1/n)$ is an appropriately chosen constant (see (Lehmann & Romano, 2005; Karwa & Vadhan, 2018; Keener, 2010) for more details and discussion). We can hence assume that $\sigma^2$ is known and proceed to show sample and space complexity bounds. [2]

For any $q \in (0, 1)$, we denote the $q$-quantile of the sample $S = (X_1, X_2, \ldots, X_n)$ as $Q_S^q$ and the $q$-quantile of the distribution as $Q_{\mathcal{D}}^q$. With this notation, for some $\beta \in (0, 1]$ and $\alpha > 0$, we wish to obtain a DP $q$-quantile $\tilde{Q}_S^q$ such that

$$\mathbb{P}[\|Q_{\mathcal{D}}^q - \tilde{Q}_S^q\| \geq \alpha] \leq \beta.$$

We shall proceed in a three-step approach: (1) Estimate a DP range of the population in sub-linear space; (2) Use this range of the population to construct a DP histogram using the stream $S$; (3) Use the DP histogram to estimate one or more quantiles via the sub-linear data structure of Greenwald and Khanna. Our main formal result for algorithm 2 can be summarized as follows (please see appendix B.2 for a complete proof).

**Theorem 4.2.** *For the one-dimensional normal distribution $\mathcal{N}(\mu, \sigma^2)$, let $S = (X_1, X_2, \ldots, X_n)$ be a data stream through which we wish to obtain $\tilde{Q}_S^q$, a DP estimate of the $q$-quantile of the distribution.*

*For any $q \in (0, 1)$, there exists an $(\epsilon, \delta)$-DP algorithm such that, with probability at least $1 - \beta$, we obtain $|Q_{\mathcal{D}}^q - \tilde{Q}_S^q| \leq \alpha$ for any $\alpha > 0$, $\beta \in (0, 1], \epsilon, \delta \in (0, 1/n)$ and for stream length*

$$n \geq \max\left\{\min\left\{A, B\right\}, C\right\}, \quad where$$

$$A = O\left(\frac{R}{\epsilon\sigma\alpha}\log\frac{R}{\sigma\beta}\right), B = O\left(\frac{R}{\epsilon\sigma\alpha}\log\frac{1}{\beta\delta}\right), C = O\left(\frac{R^2}{\sigma^2\alpha^2}\log\frac{1}{\beta}\right)$$

*as long as $\mu \in (-R, R)$ and using space of $O(\max\{\frac{R}{\sigma}, \frac{1}{\alpha}\log\alpha n\})$.*

In the case where $d = 1$, by Theorem 4.2, there exists an $(\epsilon, \delta)$-DP algorithm $\tilde{Q}_S^q$ such that if $\mu \in (-R, R)$ then using space of $O(\max\{\frac{R}{\sigma}, \frac{1}{\alpha}\log\alpha n\})$ (with probability 1) as long as the stream length $n$ is at least

$$\Omega\left(\max\left\{\min\left\{O\left(\frac{R}{\epsilon\sigma\alpha}\log\frac{R}{\sigma\beta}\right), O\left(\frac{R}{\epsilon\sigma\alpha}\log\frac{1}{\beta\delta}\right)\right\}, O\left(\frac{R^2}{\sigma^2\alpha^2}\log\frac{1}{\beta}\right)\right\}\right),$$

we get the guarantee that, for all $\beta \in (0, 1], \alpha > 0$, $\mathbb{P}[\|Q_{\mathcal{D}}^q - \tilde{Q}_S^q\| \geq \alpha] \leq \beta$. Intuitively, this means that: (1) **Space**: We need less space to estimate any quantile with DP guarantees if the distribution is less concentrated (i.e., $\sigma$ can be large) or if we do not require a high degree of accuracy for our queries (i.e., $\alpha$ can be large). (2) **Stream Length**: We need a large stream length to estimate quantiles if we require a high degree of accuracy (i.e., smaller $\beta, \alpha$), or do not have a good public estimate of $\mu$ (large $R$), or have small privacy parameters (small $\epsilon, \delta$), or have concentrated datasets (small $\sigma$).

## 5 Continual Observation

We now describe how our one-shot approach can be used as a black box to obtain a continual observation solution (Dwork et al., 2010; Chan et al., 2011). In the absence of privacy, we recall that an $\alpha/2$-approximate solution for the $q$-quantile problem on a stream of length $n$ allows for an additive error of $\alpha n/2$ in the rank of any candidate $q$-quantile solution. It follows that if we append any arbitrary $\alpha n/2$-many elements to the end

---

[2]One could also estimate the variance in a DP way and then prove the complexity bounds.

---

**Algorithm 3:** Continual Observation DP Quantiles

**Data:** Input stream $X = (x_1, \ldots, x_n)$ for some $n > n_{\min} = \Omega\left(\frac{1}{\alpha^2\epsilon} \log n \log\left(\frac{|\mathcal{X}| \log n}{\alpha\beta}\right)\right)$, privacy
parameter $\epsilon$, approximation parameter $\alpha$, target quantile $q$

**1** , failure probability $\beta$ Let $s = 0$
**2** $\epsilon^* \leftarrow O(\alpha\epsilon / \log n)$
**3** $\alpha^* \leftarrow \alpha/2$
**4** $\beta^* \leftarrow \frac{\log n}{\alpha\beta}$
**5** $\mathrm{cp}(s) \leftarrow n_{\min}$ /* stores stream checkpoints                              */
**6** Instantiate $\mathrm{GK} \leftarrow GK(\alpha^*)$
**7** $v_s \leftarrow \perp$ /* holds $\alpha$-approximate $q$-quantile                              */
**8** **for** *stream element $x_s \in X$* **do**
**9**     $s \leftarrow s + 1$
**10**     $\mathrm{GK}.insert(x_s)$
**11**     **if** $s = \lceil \mathrm{cp}(s-1) \rceil$ **then**
**12**        $v_s \leftarrow \mathtt{DPExpGK}(\mathrm{GK}, \epsilon^*, \alpha^*, q, \beta^*)$
**13**        $\mathrm{cp}(s) \leftarrow \mathrm{cp}(s-1)(1 + \alpha/2)$
**14**     **else**
**15**        $v_s \leftarrow v_{s-1}$
**16**        $\mathrm{cp}(s) \leftarrow \mathrm{cp}(s-1)$

---

of the stream, these new elements can lead to an additional additive error of at most $\alpha n/2$. This suggests that on releasing a DP quantile estimate at any point in the stream, we can avoid updating our $q$-quantile estimate for a proportionate number of additional elements whilst maintaining a $(1+\alpha)$-approximation. This is essentially the idea behind the *flip number* used in the study of adversarially robust streaming algorithms. At a high level, the flip number is a measure of the number of times the output of an online algorithm changes by more than a $(1 \pm \alpha)$ multiplicative factor (see (Ben-Eliezer et al., 2020) for more detail).

Building on this idea, we can show that we only need update our estimate after every additional $\alpha/2$-fraction as many elements are added as were present at the previous estimate. At the beginning of this process, i.e. when no stream elements have been processed, for the first few elements ($n_{\min}$-many, as described in the pseudo-code) we will need to update our online estimate every time or omit producing any estimate. However, after this short warm-up prefix of the stream, a relatively small number of estimates serve as $(1 + \alpha)$-approximate $q$ quantiles for every point in the remainder of the stream.

Accounting for the additional requirement of privacy is now easy - we merely need to privatize the estimate at each of the *checkpoints*, which are the points in the stream where the quantile estimate must be updated. For $n$ elements in all there are $O\left(\log_{1+\alpha} \frac{n}{n_{\min}}\right)$ many such elements, which for a suitable choice of $n_{min}$ simplifies essentially to $O\left(\frac{\log n}{\epsilon\alpha}\right)$. In other words we see that the privacy loss scales only with the logarithm of the length of the stream because of the relatively few checkpoints that occur. The formal statement is as below, and we present a complete proof in appendix B.3.

**Theorem 5.1.** *Let $\epsilon, \alpha > 0$, $n \in \mathbb{Z}$. For any $\beta \in (0, 1]$, with probability $\geq 1 - \beta$, Algorithm 3 maintains an $\alpha$-approximate $q$-quantile at every point $s$ in the data stream for $s = \Omega\left(\frac{\log n \log |\mathcal{X}|/\beta}{\alpha^2\epsilon}\right)$. Furthermore, Algorithm 3 satisfies $\epsilon$-DP and has space complexity $\Omega\left(\frac{1}{\alpha^2\epsilon} \log^2 n \log\left(\frac{|\mathcal{X}| \log n}{\alpha\beta}\right)\right)$.*

We conclude by mentioning that our continual observation solution incurs roughly a $O(\log n/\alpha)$ overhead in the space complexity, which is in line with classical works in differential privacy and adversarially robust streaming (Ben-Eliezer et al., 2020; Dwork et al., 2010). Concurrently, Stemmer and Kaplan (Kaplan & Stemmer, 2021) developed a notion of *streaming sanitizers* which yields a continual observation guarantee "for free", without incurring such an overhead over the one-shot case.

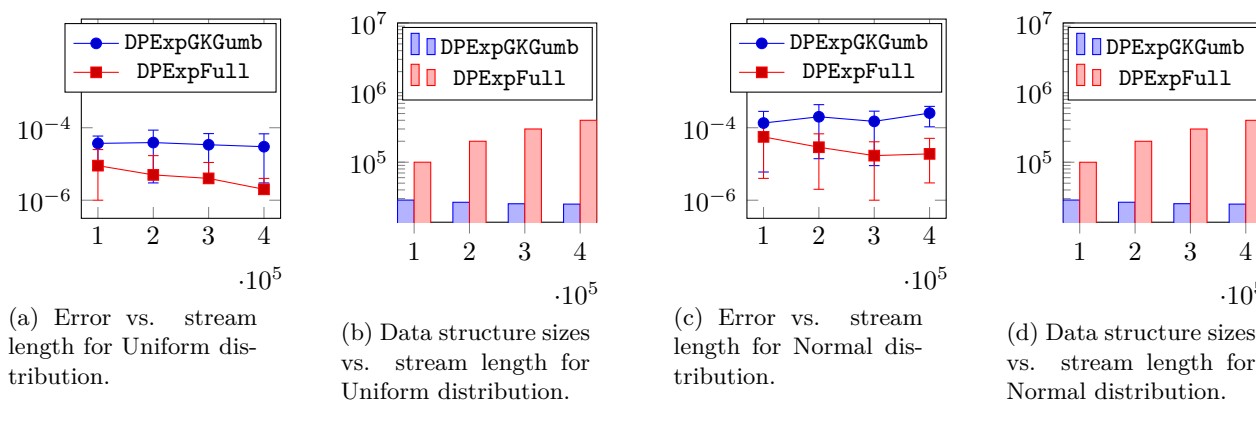

Figure 2: `DPExpGKGumb` versus `DPExpFull` for uniform or normally distributed data, $\alpha = 10^{-4}$, $\epsilon = 1$, $q = 0.5$

## 6 Experimental Evaluation

In this section, we experimentally evaluate our sublinear-space exponential-based mechanism, `DPExpGKGumb`. We study how well our algorithm performs in terms of accuracy and space usage. This section validates the theoretical results in the prequel - we find that the space complexity of the algorithm is indeed very small in practice, and that the accuracy is typically closely tied to the approximation parameter $\alpha$. Our main baselines will be `DPExpFull`, which applies the exponential mechanism on the full data set without any sketching algorithms (see appendix C.1 for further implementation details), and the true quantile value. The true quantile values are used to compute *relative error*, the absolute value of the difference between the estimated quantile and the true quantile, divided by the standard deviation of the data set. We graph the mean relative error over 100 trials of the exponential mechanism per experiment, as well as the 80% confidence interval computed by taking the 10th and 90th percentile. We fix the target quantile to be estimated to $q = 0.5$ (the median) throughout as the relative error and the space usage generally do not seem to vary much with choice of $q$.

### 6.1 Synthetic data sets

In this subsection, we compare our methods on synthetically generated datasets. We vary the parameters of the **Stream Length** $n$ (the size of the input stream $x_1, \ldots, x_n$), the **Approximation Parameter** $\alpha$ (the approximation factor used by the internal GK sketch), and the **Data Distribution**. We generate data from uniform and Gaussian distributions; we use a uniform distribution in range $[0, 1]$ (i.e., $U(0, 1)$) or a normal distribution with mean 0 and variance 1 (i.e., $N(0, 1)$), clipped to the interval $[-10, 10]$.[3]

We show results on space usage and relative error (both plotted mostly on logarithmic scales) from non-DP estimates, as we vary the parameters listed above.

In Figure 2, we vary the stream length $n$ for an approximation factor of $\alpha = 10^{-4}$. The streams are either normally or uniformly distributed. In Figures 2a and 2b, we compare `DPExpGKGumb` (space strongly sublinear in $n$) vs. `DPExpFull` (uses space of $O(n)$) in terms of space usage and accuracy. In general we find that although our method `DPExpGKGumb` incurs higher error, in absolute terms it remains quite small and the 95% confidence intervals tend to be adjacent for `DPExpGKGumb` and `DPExpFull`. However, there is a clear trend of an exponential gap developing between their respective space usages which is a natural consequence of the space complexity guarantee of the GK sketch. This holds for both distributions studied.

---

[3]Clipping is required for the exponential mechanism, as it must operate on some bounded interval of values. In any case, we never expect to see samples from $N(0, 1)$ that lie outside $[-10, 10]$ for any practical purpose; the probability for any given sample to satisfy this is minuscule, at about $\approx 10^{-21}$.

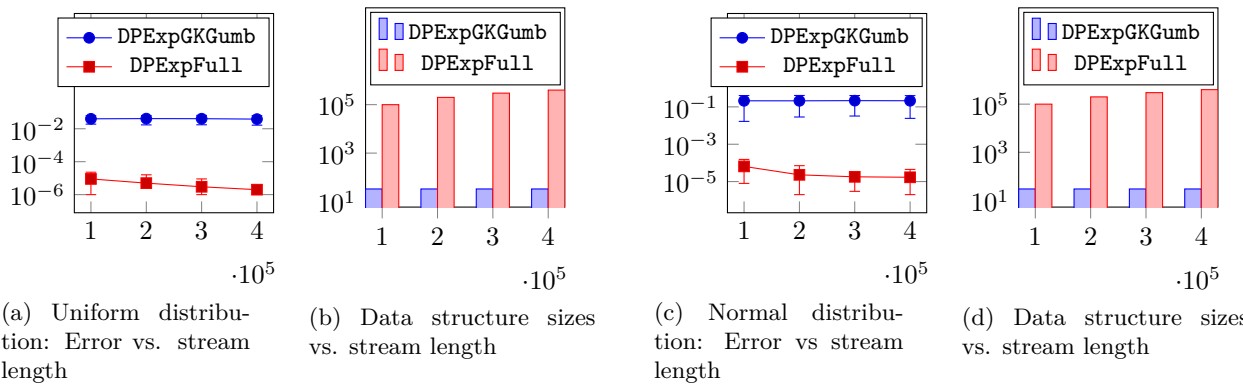

Figure 3: `DPExpGKGumb` versus `DPExpFull` for uniform or normally distributed data, $\alpha = 10^{-1}$, $\epsilon = 1$, $q = 0.5$

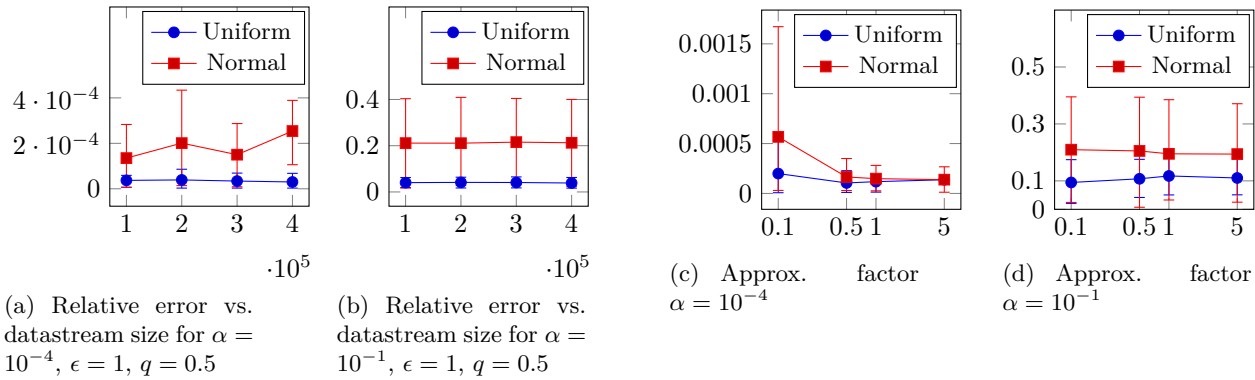

Figure 4: Relative error versus approximation factor.

In Figure 3, we vary the stream length for a relatively large approximation factor of 0.1. Here we see that compared to the non-approximate method we incur far higher error, although there is also a concomitant increase in the space savings. This is not a typical use-case since the non-private error can itself be large, but we get a complete picture of how space usage and performance vary with this user-defined parameter.

In Figures 4c and 4d, we vary the approximation factor. In the small approximation setting we see the inverse tendency of accuracy with privacy which is characteristic of most DP algorithms. However, in the large approximation setting, there is no such clear drop in performance with more privacy. This motivates the question of determining the true interplay between the approximation factor $\alpha$ and the private parameter $\epsilon$, as discussed further in Section 7.

Our algorithm performs well in practical settings where one wishes to estimate some quantity across all data items privately and using small space. For example, our results indicate that choosing an approximation factor of $\alpha = 10^{-4}$ induces an error which is also of order about $10^{-4}$ for privately computing parameters chosen according to a uniform or normal distribution, all while saving orders of magnitude in the space complexity.

## 6.2 Real-World Datasets

We repeat our investigation of the utility and space complexity comparison between `DPExpGKGumb` and `DPExpFull` with the following real-world data sets (Dua & Graff, 2017): (1) **Taxi Service Trajectory**: A dataset from the UCI machine learning repository describing trajectories performed by all 442 taxis (at

---

[4]The 80% CI for error incurred by `DPExpFull` was entirely supported on the point 0 and drops off axis as we use the log scale.

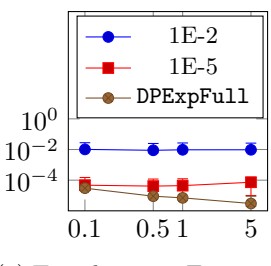
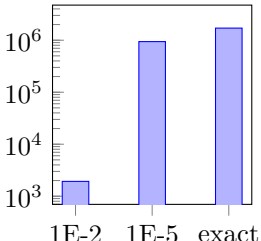
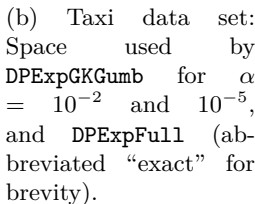
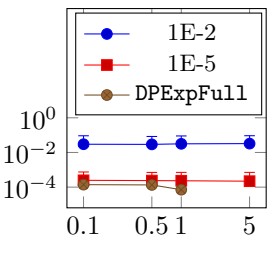
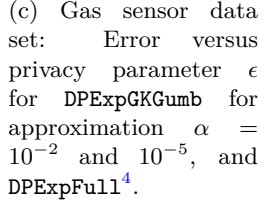
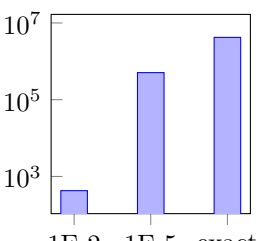

(a) Taxi data set: Error versus privacy parameter $\epsilon$ for DPExpGKGumb for approximation $\alpha = 10^{-2}$ and $10^{-5}$, and DPExpFull.

(b) Taxi data set: Space used by DPExpGKGumb for $\alpha = 10^{-2}$ and $10^{-5}$, and DPExpFull (abbreviated "exact" for brevity).

(c) Gas sensor data set: Error versus privacy parameter $\epsilon$ for DPExpGKGumb for approximation $\alpha = 10^{-2}$ and $10^{-5}$, and DPExpFull[4].

(d) Gas sensor data set: Space used by DPExpGKGumb for $\alpha = 10^{-2}$ and $10^{-5}$, and DPExpFull (abbreviated "exact" for brevity).

Figure 5: Taxi and Gas sensor data sets.

the time) in the city of Porto in Portugal (Moreira-Matias et al., 2013). This dataset contains real-valued attributes with about 1.5 million instances. (2) **Gas Sensor Dataset**: A UCI repository dataset containing recordings of 16 chemical sensors exposed to varying concentrations of two gas mixtures (Fonollosa et al., 2015). The sensor measurements are acquired continuously during a 12-hour time range and contains about 4 millions instances. We pick a real-valued attribute from each dataset (the TIMESTAMP and the first ETHYLENE_CO gas sensor value, respectively) and calculate the median on these datasets. The results are reported in Figures 5a and 5b, and Figures 5c and 5d respectively.

We see that the larger the approximation factor, the larger the space savings are with DPExpGK. Comparing DPExpFull to DPExpGK, we see space savings of 2 times up to 1000 times as we vary the approximation factor. These results are consistent with our expectations that the space savings are inversely proportional to the allowed approximation factor.

# 7 Conclusion & Future Work

In this work, we presented sublinear-space and differentially private algorithms for approximately estimating quantiles in a dataset. Our solutions are two-part: one based on the exponential mechanism and efficiently implemented via the use of the Gumbel distribution; the other based on constructing histograms. Our algorithms are supplemented with theoretical utility guarantees. Furthermore, we experimentally validate our methods on both synthetic and real-world datasets. Our work leaves room for further exploration in various directions.

**Interplay between $\alpha$ and $\epsilon$:** The space complexity bounds we obtain are (up to lower order terms) inversely linear in $\alpha$ and in $\epsilon$. While it is either known or easy to show that such linear dependence in each of these parameters in itself is necessary, it is not clear whether the $\alpha^{-1}\epsilon^{-1}$ term in Theorem 4.1 can be replaced with, say, $\alpha^{-1} + \epsilon^{-1}$. Such an improvement, if possible, seems to require substantially modifying the baseline Greenwald-Khanna sketch or adding randomness.

**Alternative streaming baselines:** We base our mechanisms upon the GK-sketch, which is known to be space-optimal among deterministic streaming algorithms for quantile approximation. The use of a deterministic baseline simplifies the analysis and the overall solution, but better randomized streaming algorithms for the same problem are known to exist. What would be the benefit of working, e.g., with the (optimal among randomized algorithms) KLL-sketch (Karnin et al., 2016)?

**Dependence in universe size:** The dependence of our space complexity bounds in the size of the universe, $\mathcal{X}$, is logarithmic. Recent work of Kaplan et al. (Kaplan et al., 2020) (see also (Bun et al., 2015)) on

the sample (not space) complexity of privately learning thresholds in one dimension, a fundamental problem at the intersection of learning theory and privacy, demonstrate a bound polynomial in $\log^* |\mathcal{X}|$ on the sample complexity. As quantile estimation and threshold learning are closely related problems, this raises the question of whether techniques developed in the aforementioned papers can improve the dependence on $|\mathcal{X}|$ in our bounds.

**Random order:** The results presented here (except for those about normally distributed data) all assume that the data stream is presented in worst case order, an assumption that may be too strong for some scenarios. Can improved bounds be proved when the data elements are chosen in advance but their order is chosen randomly? This can serve as a middle ground between the most general case (which we address in this paper) and the case where data is assumed to be generated according to a certain distribution.

## 8 Acknowledgements

D.A. was supported by a Junior Fellowship from the Simons Foundation Society of Fellows, Cooperative Agreement CB20ADR0160001 with the U.S. Census Bureau, and a Fellowship from Meta AI. Most of this work was done while he was a Ph.D. student at Harvard University. A.C. was supported in part by NSF CAREER grant 1750716.

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

## A   Greenwald-Khanna Sketch

For completeness of our algorithm's description, we specify the operations in the Greenwald-Khanna (GK) non-private sketch. Throughout, we will use $n = n(t)$ to denote the number of elements encountered up to time $t \in \mathbb{Z}_+$. Some of the operations outlined here will be used a subroutines for the DP procedures.

### A.1   The Sketch

Let $X = (x_1, x_2, \ldots, x_n)$ be a stream of items and $S(X)$ be the resulting sketch with size sublinear in $n$. The GK sketch stores

$$S(X) = \langle t_0, t_1, \ldots, t_{s-1} \rangle, \quad \forall i \in \{0, \ldots, s-1\}, t_i = (v_i, g_i, \Delta_i),$$

where $g_i = r_{min}(v_i) - r_{min}(v_{i-1})$ and $\Delta_i = r_{max}(v_i) - r_{min}(v_i)$. We reserve $v_0, v_{s-1}$ be denote the smallest and largest elements seem in the stream $X$, respectively. We use $S(X)[i]$ to refer to the $i$th tuple in the sketch $S(X)$. i.e., for any $i$, $S(X)[i] = t_i = (v_i, g_i, \Delta_i)$.

Implicitly, the goal is to (implicitly) maintain bounds $r_{min}(v)$ and $r_{max}(v)$ for every $v$ in $S(X)$. $r_{min}(v)$ and $r_{max}(v)$ are the lower and upper bounds on the rank of $v$ amongst all items in $X$, respectively. We can compute these bounds as follows:

$$r_{min}(v_i) = \sum_{j \leq i} g_j, \quad r_{min}(v_i) = \sum_{j \leq i} g_j + \Delta_i.$$

As a result, $g_i + \Delta_i - 1$ is an upper bound on the number of items between $v_{i-1}$ and $v_i$. In addition, $n = \sum_i g_i$.

The sketch is built in such a way to guarantee (maximum) error of $\max_{i=0}^{s-1}(g_i + \Delta_i)/2$ for approximately computing any quantile using the sketch.

We will also impose a tree structure over tuples in $S(X)$ (mostly because of the merge procedure) as follows: the tree $T(X)$ associated with $S(X)$ has a node $V_i$ for each $t_i$. The parent of a node $V_i$ is the node $V_j$ such that $j$ is the smallest index greater than $i$ with $\text{band}(t_j) > \text{band}(t_i)$.

$\text{band}(t_i)$ is the band of $\Delta_i$ at time $n$ and $\text{band}_\tau(n)$ as all tuples that had band value of $\tau$. All possible values of $\Delta$ are denoted as bands and it can take on values between
$(0, \frac{1}{2}2\alpha n, \frac{3}{4}2\alpha n, \ldots, \frac{2^i-1}{2^i}2\alpha n, \ldots, 2\alpha n - 1, 2\alpha n)$ corresponding to capacities of $(2\alpha n, \alpha n, \ldots, 8, 4, 2, 1)$.

### A.2 Quantile

Algorithm 4 computes the $\alpha$-approximate $q$-quantile based on the sketch $S(X)$ that has size that is sublinear in $n$.

The algorithm goes through all tuples and checks if the condition $\max(r - r_{min}(v_i), r_{max}(v_i) - r) \leq \alpha n$ is satisfied and return $(i, v_i)$ as the representative approximate quantile. This algorithm will be used a subroutine for one or more of our differentially private algorithms.

**Lemma A.1** (Proposition 1 & Corollary 1 (Greenwald & Khanna, 2001))**.** *If after receiving $n$ items in the stream, the sketch $S(X)$ satisfies the property $\max_i(g_i + \Delta_i) \leq 2\alpha n$, then Algorithm 4 returns an $\alpha$-approximate $q$-quantile.*

*Proof.* The algorithm computes $r = \lceil qn \rceil$. Then the condition $\max(r - r_{min}(v_i), r_{max}(v_i) - r) \leq \alpha n$ clearly is (by definition) an $\alpha$-approximate $q$-quantile. We still need to show that such $v_i$ always exists. First set $e = \max_i(g_i + \Delta_i)/2$. If $r > n - e$, then $r_{min}(v_{s-1}) = r_{max}(v_{s-1}) = n$ so that $i = s - 1$ satisfies the property. When $r \leq n - e$, then the algorithm chooses the smallest index $j$ such that $r_{max}(v_j) > r + e$ so that $r - e \leq r_{min}(v_{j-1})$. This follows since if $r - e > r_{min}(v_{j-1})$ then $r_{max}(v_j) = r_{min}(v_{j-1}) + g_j + \Delta_j > r_{min}(v_{j-1}) + 2e$ which contradicts the definition of $e$. $\square$

---

**Algorithm 4:** $\text{Quantile}(S(X), q, n, \alpha)$: Computing $\alpha$-Approximate Quantiles

**Input:** $S(X), q, n, \alpha$ (approximation parameter)
**1** Compute $r = \lceil qn \rceil$
**2 for** $i = 0, \ldots, s-1$ **do**
**3** $\quad (v_i, g_i, \Delta_i) = S(X)[i]$
**4** $\quad$ **if** $\max(r - r_{min}(v_i), r_{max}(v_i) - r) \leq \alpha n$ **then**
**5** $\quad\quad$ **return** $(i, v_i)$

**6 return** $\perp$

---

### A.3 Insert

Algorithm 5 goes through a stream of items and inserts into the sketch. The algorithm calls a compress operator on the data structure every time that $i \equiv 0 \mod \frac{1}{2\alpha}$ for any $i \in [m]$.

Algorithm 6 inserts a particular item $x_n$ into the data structure $S(X)$. In the special case where $x_n$ is a minimum or maximum, it inserts the tuple $(x_n, 1, 0)$ at the beginning or end of $S(X)$. Otherwise, it finds an index $i$ such that $v_{i-1} \leq x_n < v_i$ and then inserts the tuple $(x_n, 1, \lfloor 2\alpha n \rfloor)$ into $S(X)$ at position $i$.

### A.4 Compress

The Compress operation is an internal operation used for compressing (contiguous) tuples in $S(X)$. The goal of this operation is to merge a node and its descendants into either its right sibling or parent node. After merge, we have to maintain the property that the tuple is not full. A tuple is full when $g_i + \Delta_i \geq \lfloor 2\alpha n \rfloor$.

---

**Algorithm 5:** Inserting a stream of items into Summary Sketch

---

**Data:** $x_1, x_2, \ldots, x_m, \ldots$
**Input:** $S(X), \alpha$ (approximation parameter)

**1** $n = 0$
**2** **for** $i = 1, \ldots, m, \ldots$ **do**
**3**      **if** $i \equiv 0 \mod \frac{1}{2\alpha}$ **then**
**4**          Compress($S(X), \alpha, n$)
**5**      Insert($S(X), \alpha, x_i$)
**6**      $n = n + 1$
**7** **return** $S(X), n$

---

**Algorithm 6:** Insert($S(X), \alpha, x_n$): Inserting into Summary Sketch

---

**Data:** $x_n$
**Input:** $S(X), \alpha$ (approximation parameter)

**1** $(v_0, g_0, \Delta_0) = S(X)[0]$
**2** $(v_{s-1}, g_{s-1}, \Delta_{s-1}) = S(X)[s-1]$
**3** **if** $x_n < v_0$ **then**
**4**      Shift all positions in $S(X)[0 \ldots s-1]$ to $S(X)[i \ldots s]$
**5**      $S(X)[0] = (x_n, 1, 0)$
**6** **else if** $x_n > v_{s-1}$ **then**
**7**      $S(X)[s] = (x_n, 1, 0)$
**8** **else**
**9**      **for** $i = 0, \ldots, s-1$ **do**
**10**          $(v_i, g_i, \Delta_i) = S(X)[i]$
**11**          **if** $v_{i-1} \leq x_n < v_i$ **then**
**12**              Shift all positions in $S(X)[i \ldots s-1]$ to $S(X)[i+1 \ldots s]$
**13**              $S(X)[i] = (x_n, 1, \lfloor 2\alpha n \rfloor)$
**14** **return** $S(X), s + 1$

---

**Algorithm 7:** Compressing the Sketch

---

**Input:** $S(X), \alpha$ (approximation parameter)$, n$

**1** **if** $n < \frac{1}{2\alpha}$ **then**
**2**      **return**
**3** **for** $i = s - 2, \ldots, 0$ **do**
**4**      $t_i = (v_i, g_i, \Delta_i) = S(X)[i]$
**5**      $t_{i+1} = (v_{i+1}, g_{i+1}, \Delta_{i+1}) = S(X)[i+1]$
**6**      Compute $g_i^*$, the sum of $g$-values of tuple $t_i$ and its descendants
**7**      **if** $band(t_i) \leq band(t_{i+1})$ & $(g_i^* + g_{i+1} + \Delta_{i+1} < 2\alpha n)$ **then**
**8**          Delete all descendants of $t_i$ and the tuple $t_i$ from sketch $S(X)$
**9**          Update $t_{i+1}$ in $S(X)$ to $(v_{i+1}, g_i^* + g_{i+1}, \Delta_{i+1})$

---

By Proposition A.2, a node and its children will form a contiguous segment. Let $g_i^*$ be the sum of $g$-values of tuple $t_i$ and all of its descendants. Then merging $t_i$ and its descendants would update $t_{i+1}$ in $S(X)$ to $(v_{i+1}, g_i^* + g_{i+1}, \Delta_{i+1})$ and delete $t_i$ and all of its descendants.

**Proposition A.2** (Proposition 4 in (Greenwald & Khanna, 2001)). *For any node $V$, the set of all its descendants in the tree forms a contiguous segment in $S(X)$.*

### A.5   Formal guarantee

We briefly recall the main guarantees of the GK sketch that we appeal to in the main body of this work and provide a proof for the statements that are not reproduced from previous work.

**Lemma 3.7.** *Given a data stream $x_1, \ldots, x_n$ of elements drawn from $\mathcal{X}$, there exists a sketching algorithm that outputs a list $S(X)$ of $s = O\left(\frac{1}{\alpha} \log \alpha n\right)$ many tuples $(v_i, g_i, \Delta_i) \in (\mathcal{X} \times \mathbb{N}) \times \mathbb{N} \times \mathbb{N}$ for $i = 1, \ldots, s$ such that if $X$ is the data multi-set then:*

*1. $\operatorname{rank}(X, \operatorname{val}(v_i)) \subset [\sum_{j \leq i} g_j, \Delta_i + \sum_{j \leq i} g_j]$.*

*2. $g_i + \Delta_i \leq 2\alpha n$.*

*3. The first tuple is $(\min\{x \in X\}, 1, 0)$ and the last tuple is $(\max\{x \in X\}, 1, 0)$.*

*4. The $v_i$ are sorted in ascending order. Without loss of generality, the lower interval bounds $\sum_{j \leq i} g_j$ and upper interval bounds $\Delta_i + \sum_{j \leq i} g_j$ are also sorted in increasing order.*

*Proof.* The first three statements are part of the GK sketch guarantee. For the third statement, i.e., to see that the $v_i$ are sorted in ascending order, we see that the GK sketch construction ensures that $\operatorname{val}(v_i) \leq \operatorname{val}(v_{i+1})$ for all $i$. Since an insertion operation always inserts a repeated value after all previous occurrences and the tuple order is always preserved, it follows that $\operatorname{ix}(v_i) \leq \operatorname{ix}(v_{i+1})$ as well, so in sum $\operatorname{rank}(X, v_i) \leq \operatorname{rank}(X, v_{i+1})$. In other words, the sort order in the GK sketch is *stable*.

The fact that the sequence $\sum_{j \leq i} g_j$ is sorted in increasing order follows from the non-negativity of the $g_i$. To ensure that $\Delta_i + \sum_{j \leq i} g_j$ are sorted in increasing order note that we always have that $\operatorname{rank}(X, v_i) \leq \operatorname{rank}(X, v_{i+1})$ so that we can decrement $\Delta_i$ and ensure that $\Delta_i + \sum_{j \leq i} g_j \leq \Delta_{i+1} + \sum_{j \leq i+1} g_j$ without violating the guarantees of the GK sketch. $\qquad \square$

## B   Omitted proofs

We first recall and add some definitions that we will require in the sequel.

**Definition B.1.** Let $X = ((x_1, 1), \ldots, (x_n, n))$ (sometimes implicitly referred to as $X = (x_1, \ldots, x_n)$) be a stream of elements drawn from some finite totally ordered data universe $\mathcal{X}$, i.e., $x_i \in \mathcal{X}$ for all $i \in [n]$.

1. **Rank**: Given a totally ordered finite data universe $\mathcal{X}$, a data set $X$ and a value $x \in \mathcal{X}$, let $\operatorname{rank}_X(x) = \operatorname{rank}(X, x) = \sum_{y \in X} \mathbb{1}[y \leq x]$.

2. For $(x_i, i) \in X$, let $\operatorname{val}((x_i, i)) = x_i$ and $\operatorname{ix}((x_i, i)) = \sum_{j \leq i} |\{(x_j, j) : x_j < x_i \text{ or } x_j = x_i, j < i\}|$.

3. For $v_1, v_2 \in X$, we say that $v_1 \leq v_2$ if $\operatorname{ix}(v_1) \leq \operatorname{ix}(v_2)$.

4. The $q$-quantile of $X$ is $\operatorname{val}(v)$ for $v \in X$ with $\operatorname{ix}(v) = \lceil qn \rceil$.

5. For $x \in \mathcal{X}$, we define $r_{\min}(x) = |\{v \in X : \operatorname{val}(v) < x\}|$, $r_{\max}(x) = |\{v \in X : \operatorname{val}(v) \leq x\}|$ and $\operatorname{rank}(X, x)$ to be the interval $[r_{\min}(x), r_{\max}(x)]$.

6. We say that $x \in \mathcal{X}$ is an $\alpha$-approximate $q$-quantile for $X$ if $\operatorname{rank}(X, x) \cap [\lceil qn \rceil - \alpha n, \lceil qn \rceil + \alpha n] \neq \emptyset$.

With this notation, the data set $X$ is naturally identified as a multi-set of elements drawn from $\mathcal{X}$.

**Definition B.2.** Let $\hat{r}_{\min}(x) = \max\{\sum_{j \leq i} g_j : \operatorname{val}(v_i) < x\}$ and $\hat{r}_{\max}(x) = \min\{\Delta_i + \sum_{j \leq i} g_j : \operatorname{val}(v_i) > x\}$. Note that for every $v \in X$ such that $\operatorname{val}(v) = x$, $\operatorname{ix}(v) \in [\hat{r}_{\min}(x), \hat{r}_{max}(x)]$.

### B.1 Proofs of privacy and utility for `DPExpGK`

**Outline:** From Definition B.2 to Lemma B.5, we formalize how the GK sketch may be used to construct rank interval estimates for any data domain value. We then recall and apply the exponential mechanism with a score function derived from the GK sketch (Definition B.6 to Lemma B.9 and Algorithm 1), and derive the error guarantee Lemma B.11. We conclude this subsection with a detailed description of an efficient implementation of the exponential mechanism (Algorithm 8 and Lemma B.12), and summarize our final accuracy and space complexity guarantees in Theorem 4.1.

*Remark* B.3. We can add two additional tuples $(-\infty, 0, 0)$ and $(\infty, 0, 0)$ to the sketch, which corresponds to respective rank intervals $[0, 0]$ and $[n + 1, n + 1]$. The bounds $g_i + \Delta_i \leq 2\alpha n$ are preserved. This will ensure that the sets $\{i : \text{val}(v_i) < x\}$ and $\{i : \text{val}(v_i) > x\}$ for any $x \in \mathcal{X}$ are always non-empty.

We formalize the rank interval estimation in a partition-wise manner as below.

**Lemma B.4.** *Given a GK sketch* $(v_1, g_1, \Delta_1), \ldots, (v_s, g_s, \Delta_s)$, *for every* $x \in \mathcal{X}$ *one of the following two cases holds:*

*1.* $x = v_i$ *for some* $i \in [s]$ *and* $i = \min\{j : v_j = x\}$,

$$\hat{r}_{\min}(x) = \sum_{j \leq i} g_j$$

$$\hat{r}_{\max}(x) = \min\{\Delta_{i^*} + \sum_{j \leq i^*} g_j : \exists i^*, \text{val}(v_{i^*}) > \text{val}(v_i)\}$$

*2.* $x \in (v_{i-1}, v_i)$, *i.e.,* $x > v_{i-1}$ *and* $x < v_i$ *for some* $i \in [s]$,

$$\hat{r}_{\min}(x) = \sum_{j \leq i} g_j$$

$$\hat{r}_{\max}(x) = \Delta_{i+1} + \sum_{j \leq i+1} g_j$$

*Proof.* Recall, by Remark B.3, that the first tuple and the last tuple are formal elements at $-\infty$ and $\infty$, ensuring that every data universe element either explicitly occurs in the GK sketch or lies between two values that occur in the GK sketch. Both statements now follow directly from Definition B.2 and the fact that the values $v_i$ occur in increasing order in the sketch (Lemma 3.7). $\square$

We bound the quality of the rank interval estimate $[\hat{r}_{\min}(x), \hat{r}_{\max}(x)]$ compared to the true rank interval $[r_{\min}(x), r_{\max}(x)]$ as follows.

**Lemma B.5.** $|r_{\min}(x) - \hat{r}_{\min}(x)| \leq 2\alpha n$ *and* $|r_{\max}(x) - \hat{r}_{\max}(x)| \leq 2\alpha n$.

*Proof.* Let $i^* = \text{argmax}_{i:\text{val}(v_i)<x} \sum_{j \leq i} g_j$. Then by Definition of $i^*$, we have that $\text{val}(v_{i^*}) < x \leq \text{val}(v_{i^*+1})$. It follows that

$$[\text{ix}(v_{i^*}), \text{ix}(v_{i^*+1})] \subset [\sum_{j \leq i^*} g_j, \Delta_{i^*+1} + g_{i^*+1} + \sum_{j \leq i^*} g_j]$$

$$\subset [\hat{r}_{\min}(x), \Delta_{i^*+1} + g_{i^*+1} + \hat{r}_{\min}(x)]$$

Since $r_{\min}(x) \in [\text{ix}(v_{i^*}), \text{ix}(v_{i^*+1})]$ and $g_{i^*+1} + \Delta_{i^*+1} \leq 2\alpha n$, it follows that $|r_{\min}(x) - \hat{r}_{\min}(x)| \leq 2\alpha n$. The other inequality follows analogously. $\square$

**Definition B.6** (Exponential Mechanism (McSherry & Talwar, 2007))**.** Let $u : \mathcal{S}^S \times \mathcal{R} \to \mathbb{R}$ be an arbitrary score function with global sensitivity $\delta_u$. For any database summary $d \in \mathcal{S}^S$ and privacy parameter $\epsilon > 0$, the exponential mechanism $\mathcal{E}_u^\epsilon : \mathcal{S}^S \to \mathcal{R}$ outputs $r \in \mathcal{R}$ with probability $\propto \exp(\frac{\epsilon \cdot u(S(X), r)}{2\delta_u})$ where

$$\delta_u = \max_{X \sim X', r} |u(S(X), r) - u(S(X'), r)|.$$

The following statement formalizes the trade-off between the privacy parameter $\epsilon$ and the tightness of the tail bound on the score attained by the exponential mechanism.

**Theorem B.7** ( (McSherry & Talwar, 2007; Smith, 2011)). *The exponential mechanism (Definition B.6) satisfies $\epsilon$-differential privacy. Further, the following tail bound on the utility (the score of the output element) holds:*

$$P\left(u(S(X), \mathcal{E}_u^\epsilon(S(X))) < \max_{r \in R} u(S(X), r) - \frac{2\delta_u(t + \ln s)}{\epsilon}\right) \le e^{-t},$$

*where $s$ is the size of the universe from which we are sampling from.*

To run the exponential mechanism using our approximate rank interval estimates, we define a score function as follows.

**Definition B.8.** Let $d(\cdot, \cdot)$ denote the $\ell_1$ metric on $\mathbb{R}$. Given a sketch $S(X)$, we define a score function on $\mathcal{X}$:

$$u(S(X), x) = -\min\{|y - \lceil qn \rceil| : y \in [\hat{r}_{\min}(x), \hat{r}_{\max}(x)]\}$$
$$= -d(\lceil qn \rceil, [\hat{r}_{\min}(x), \hat{r}_{\max}(x)])$$

The magnitude of the noise that is added in the course of the exponential mechanism depends on the sensitivity of the score function, which we bound from above as follows.

**Lemma B.9.** *For all $n > 1/\alpha$, the sensitivity of $u$ (i.e., $\delta_u$) is at most $4\alpha n + 2$ units.*

*Proof.* Fix any data set $X'$ neighbouring $X$ under swap DP and let $[r'_{\min}(\cdot), r'_{\max}(\cdot)]$ be the rank ranges with respect to $X'$ for values in $\mathcal{X}$. Let $[\hat{r'}_{\min}(\cdot), \hat{r'}_{\max}(\cdot)]$ denote the confidence interval derived from the GK sketch $S(X')$ for values in $\mathcal{X}$.

**Claim B.10.** $|r_{\min}(x) - r'_{\min}(x)| \le 2$, $|r_{\max}(x) - r'_{\max}(x)| \le 2$.

*Proof.* These bounds follow directly from the Definition of $r_{\min}$ and $r_{\max}$; under swap DP at most two elements of the stream are changed which implies that the count of the sets defining these terms changes by at most 1 unit each for a total shift of 2 units (in fact, this can be bounded by 1 unit). $\square$

We now prove the sensitivity bound.

$$u(S(X), x) = -d(\lceil qn \rceil, [\hat{r}_{\min}(x), \hat{r}_{\max}(x)])$$
$$\le -d(\lceil qn \rceil, [r_{\min}(x), r_{\max}(x)]) + 2\alpha n$$
$$\le -d(\lceil qn \rceil, [r'_{\min}(x), r'_{\max}(x)]) + 2\alpha n + 2$$
$$\le -d(\lceil qn \rceil, [\hat{r'}_{\min}(x), \hat{r'}_{\max}(x)]) + 4\alpha n + 2$$
$$\le u(S(X'), x) + 4\alpha n + 2.$$

Swapping the positions of $X$ and $X'$, we get the reverse bound to complete the sensitivity analysis. $\square$

We can now derive a high probability bound on the utility that is achieved by Algorithm 1.

**Lemma B.11.** *If $\hat{x}$ is the value returned* `DPExpGK` *then with probability $1 - \beta$,*

$$d(\lceil qn \rceil, [\hat{r}_{\min}(\hat{x}), \hat{r}_{\max}(\hat{x})]) \le 2\alpha n + \frac{2(4\alpha n + 2)\log(|\mathcal{X}|/\beta)}{\epsilon}.$$

*Proof.* By construction, Algorithm 1 is simply a call to the exponential mechanism with score function $u(S(X), \cdot)$, Since for any target $q$-quantile, $\lceil qn \rceil$ lies in $[0, n]$ it follows that there is some $i^* \in s$ such that $\lceil qn \rceil \in [\sum_{j \le i^*} g_j, \Delta_{i^*+1} + g_{i^*+1} + \sum_{j \le i^*} g_j]$. It follows that $d(\lceil qn \rceil, [r_{\min}(\text{val}(v_{i^*})), r_{\max}(\text{val}(v_{i^*}))]) \le 2\alpha n$

and that hence $\max(u(S(X), x)) \geq -2\alpha n$. If $x^*$ is the output of the exponential mechanism, then applying the utility tail bound we get that with probability $1 - \beta$,

$$u(S(X), x^*) \geq -2\alpha n - \frac{2(4\alpha n + 2)\log(|\mathcal{X}|/\beta)}{\epsilon}.$$

By definition of $u$, the desired bound follows. □

---

**Algorithm 8:** `DPExpGKGumb`: Implementing the Exponential Mechanism on $S(X)$ using the Gumbel Distribution (Optimized implementation of Algorithm 1)

---

**Data:** $X = (x_1, x_2, \ldots, x_n)$
**Input:** $\epsilon, \alpha$ (approximation parameter), $q \in [0, 1]$ (quantile parameters)

**1** Build summary sketch $S(X)$ and let $s = |S(X)|$.
**2** Let $(v_i, g_i, \Delta_i) = S(X)[i]$ for all $i \in [s]$
**3** maxIndex $= -1$
**4** maxValue $= -\infty$

    /* Iterating over tuple values $v_i$                                            */
**5** Let $i = 1$
**6** **while** $i <= s$ **do**
**7**     $\hat{r}_{\min} = \sum_{j \leq i} g_j$
**8**     $\hat{r}_{\max} = \min\{\Delta_{i^*} + \sum_{j \leq i^*} g_j : \text{val}(v_{i^*}) > \text{val}(v_i)\}$
**9**     $u_i = -\min\{|y - \lceil qn \rceil| : y \in [\hat{r}_{\min}, \hat{r}_{\max}]\}$.
**10**     $f = \frac{\epsilon}{2} u_i$
**11**     $\tilde{f} = f + \text{Gumb}(0, 1)$
**12**     **if** $\tilde{f} > maxValue$ **then**
**13**         maxIndex $= (i, \text{tuple})$
**14**         maxValue $= \tilde{f}$
**15**     $i \leftarrow \min\{j : v_j > v_i\}$

    /* Iterating over intervals between tuples $\mathcal{X}(v_{i-1}, v_i) \subset \mathcal{X}$                      */
**16** Let $i = 1$
**17** **while** $i <= s$ **do**
**18**     **if** $\mathcal{X}(v_{i-1}, v_i)$ *is not empty* **then**
**19**         $\hat{r}_{\min} = \sum_{j \leq i} g_j$
**20**         $\hat{r}_{\max} = \Delta_{i+1} + \sum_{j \leq i+1} g_j$
**21**         $u_{i-1,i} = -\min\{|y - \lceil qn \rceil| : y \in [\hat{r}_{\min}, \hat{r}_{\max}]\}$.
**22**         $f = \log(|\mathcal{X}(v_{i-1}, v_i)|) + \frac{\epsilon}{2} u_{i-1,i}$
**23**         $\tilde{f} = f + \text{Gumb}(0, 1)$
**24**         **if** $\tilde{f} > maxValue$ **then**
**25**             maxIndex $= (i, \text{interval})$
**26**             maxValue $= \tilde{f}$
**27**     $i \leftarrow i + 1$
**28** **if** $maxIndex = (i, \text{tuple})$ *for some* $i \in 1, \ldots, s$ **then**
**29**     **return** $v_i$
**30** **else if** $maxIndex = (i, \text{interval})$ *for some* $i \in 1, \ldots, s$ **then**
**31**     Pick $v \in \mathcal{X}(v_{i-1}, v_i)$ uniformly at random
**32**     **return** $v$

---

As discussed before, in general a naive implementation of the exponential mechanism as in Algorithm 1 would in general not be efficient. To resolve this issue, in Algorithm 8 we take advantage of the partition of the data domain by the score function and the *Gumbel-max* trick to implement the exponential mechanism without

any higher-order overhead and return an $\alpha$-approximate $q$ quantile. This trick has become a standard way to implement the exponential mechanism over intervals/tuples.

**Lemma B.12.** *Algorithm 8 implements Algorithm 1 on the data universe $\mathcal{X}$ with space complexity $O(|S(X)|)$ (where $S(X)$ is the GK sketch) and additional time complexity $O(|S(X)|\log|S(X)|)$.*

*Proof.* We see that it will suffice to show that Algorithm 8 executes the exponential mechanism with the same score function as Algorithm 1 to prove that it is a valid implementation of the latter.

As noted in previous work (Abernethy et al., 2017), if $Z_1, \ldots, Z_N$ are drawn i.i.d. from standard Gumbel distribution, then

$$\mathbb{P}\left[f_i + Z_i = \max_{j \in [N]}\{f_j + Z_j\}\right] = \frac{\exp(f_i)}{\sum_{j \in [N]} \exp(f_j)}, \forall i \in [N].$$

We recall that when running the exponential mechanism on $\mathcal{X}$, we want to sample the element $x \in \mathcal{X}$ with probability $\propto \exp(\epsilon u(S(X), x))$. To implement the exponential mechanism via the identification with Gumbel argmax distribution above, we will simply compute the scores $u(S(X), x)$ and let $f_i = \epsilon \cdot u(S(X), x)$.

For $x \in \mathcal{X}$ such that $x = v_i$ for some $i \in [s]$, Algorithm 8 directly computes the scores according to Definition B.8 and Lemma B.4; this is formalized by lines 5 to 15 in the pseudo code.

For $x \in \mathcal{X}$ which lie strictly between the tuple values $\{v_i : i \in [s]\}$, we proceed as follows. Fixing $i$, from Lemma B.4 we have that that for $\mathcal{X}(v_{i-1}, v_i) := \{x \in \mathcal{X} : x > v_{i-1}, x < v_i\}$, the rank confidence interval estimate is the same, i.e. $[\sum_{j \leq i-1} g_j, \Delta_i + \sum_{j \leq i} g_j]$. It follows from Definition B.8 that for all such domain values the the score function value $u(S(X), \cdot)$ is equal; this is denoted $u_{i-1,i}$ in the pseudo code. By summing the probabilities for sampling individual domain elements, it follows that the likelihood of the exponential mechanism outputting some value from the set $\mathcal{X}(v_{i-1}, v_i)$ is $\propto |\mathcal{X}(v_{i-1}, v_i)| \exp(\epsilon u_{i-1,i}/2) = \exp(\epsilon u_{i-1,i}/2 + \log(|\mathcal{X}(v_{i-1}, v_i)|))$. This is formalized by lines 16 to 27 in the pseudo code.

Finally, if some interval is selected, then by outputting elements chosen uniformly at random, we ensure that the likelihood of $x \in \mathcal{X}(v_{i-1}, v_i)$ being output is $\propto \frac{1}{|\mathcal{X}(v_{i-1}, v_i)|} \cdot \exp(\epsilon u_{i-1,i}/2 + \log(|\mathcal{X}(v_{i-1}, v_i)|)) = \exp(\epsilon u_{i-1,i})$. Note that we do not need to account for ties in the Gumbel scores as the event $f_i + Z_i = f_j + Z_j$ for any $j \neq i$ has measure 0. [5]

To bound the space and time complexity; we note that by the guarantees of the GK sketch, the size of the sketch $S(X)$ is $O((1/\alpha)\log \alpha n)$; we compute Gumbel scores by iterating over tuples and intervals of which there are at most $O(S(X))$-many of each, each computation takes at most $O(\log|S(X)|)$ time, and only the max score and index seen at any point is tracked in the course of the algorithm. $\square$

We can now state and prove our main theorem in this section, proving utility bounds for $\alpha$-approximating quantiles through `DPExpGK` with sublinear space.

**Theorem 4.1.** *Algorithm 1 is $\epsilon$-differentially private. Let $\hat{x}$ be the value returned by Algorithm 1 when initialized with target quantile $q$. The following statements hold:*

1. *Algorithm 1 can be implemented to run with space complexity $O((1/\alpha)\log \alpha n)$, such that with probability $1 - \beta$*

$$d(\lceil qn \rceil, [\hat{r}_{\min}(\hat{x}), \hat{r}_{\max}(\hat{x})]) \leq 2\alpha n + \frac{2(4\alpha n + 2)\log(|\mathcal{X}|/\beta)}{\epsilon} \quad (2)$$

2. *For $n > \frac{24\log|\mathcal{X}|/\beta}{\alpha\min\{\epsilon,1\}}$, Algorithm 1 can be implemented to run with space complexity $O\left((\alpha\epsilon)^{-1}\log(|\mathcal{X}|/\beta)\log n\right)$ such that with probability $1 - \beta$, $\hat{x}$ is an $\alpha$ approximate $q$-quantile.*

---

[5]As is usual in the privacy literature, we assume that the sampling of the $Gumb(0,1)$ distribution can be done on finite-precision computers (Balcer & Vadhan, 2018). While the problem of formally dealing with rounding has not been settled in the privacy literature (Mironov, 2012), for any practical purpose it easily suffices to store the output of the Gumbel distribution using a few computer words.

Our dependence in $\alpha$, which for practical purposes is usually the most important term, is optimal. Recent subsequent work by Kaplan and Stemmer (Kaplan & Stemmer, 2021) shows how to improve the dependence in other parameters if approximate (rather than pure) differential privacy is allowed, or if the stream length is large enough.

*Proof.* The privacy guarantee of Algorithm 1 follows from the privacy guarantee of the exponential mechanism and Lemma B.12. The accuracy bound in equation 2 is simply a restatement of Lemma B.11, and the space complexity bounds follow from the GK sketch space complexity bound $O\left(\frac{1}{\alpha}\log \alpha n\right)$. To derive the second statement, we substitute $\frac{\alpha\min\{\epsilon,1\}}{24\log(|\mathcal{X}|/\beta)}$ for the approximation parameter $\alpha$ in equation 2 and get

$$d(\lceil qn\rceil, [\hat{r}_{\min}(\hat{x}), \hat{r}_{\max}(\hat{x})])$$

$$\leq 2\cdot\frac{\alpha\min\{\epsilon,1\}n}{24\log(|\mathcal{X}|/\beta)} + \frac{2(4(\frac{\alpha\min\{\epsilon,1\}}{24\log(|\mathcal{X}|/\beta)})n + 2)\log(|\mathcal{X}|/\beta)}{\epsilon}$$

$$\leq \frac{\alpha n}{12\log(|\mathcal{X}|/\beta)} + \frac{\alpha n}{3} + \frac{4\log|\mathcal{X}|/\beta}{\epsilon}$$

$$\leq \frac{\alpha n}{12} + \frac{\alpha n}{3} + \frac{\alpha n}{3}$$

$$\leq \alpha n.$$

The space complexity bound now follows directly from the space complexity bound derived in Lemma B.12, the space complexity bound $O((1/\alpha)\log\alpha n)$ for the GK sketch, and by substituting $\frac{\alpha\min\{\epsilon,1\}}{24\log(|\mathcal{X}|/\beta)}$ for $\alpha$. Sincve the approximation parameter must be greater than $1/n$, we have that $n \geq \frac{24\log(|\mathcal{X}|/\beta)}{\alpha\min\{\epsilon,1\}}$. $\square$

## B.2 Proofs of privacy and utility for `DPHistGK`

In this section we prove Theorem 4.2, which we restate here for ease of reference.

**Theorem 4.2.** *For the one-dimensional normal distribution $\mathcal{N}(\mu, \sigma^2)$, let $S = (X_1, X_2, \ldots, X_n)$ be a data stream through which we wish to obtain $\tilde{Q}_S^q$, a DP estimate of the q-quantile of the distribution.*

*For any $q \in (0,1)$, there exists an $(\epsilon, \delta)$-DP algorithm such that, with probability at least $1 - \beta$, we obtain $|Q_{\mathcal{D}}^q - \tilde{Q}_S^q| \leq \alpha$ for any $\alpha > 0$, $\beta \in (0,1], \epsilon, \delta \in (0, 1/n)$ and for stream length*

$$n \geq \max\left\{\min\left\{A, B\right\}, C\right\}, \quad where$$

$$A = O\left(\frac{R}{\epsilon\sigma\alpha}\log\frac{R}{\sigma\beta}\right), B = O\left(\frac{R}{\epsilon\sigma\alpha}\log\frac{1}{\beta\delta}\right), C = O\left(\frac{R^2}{\sigma^2\alpha^2}\log\frac{1}{\beta}\right)$$

*as long as $\mu \in (-R, R)$ and using space of $O(\max\{\frac{R}{\sigma}, \frac{1}{\alpha}\log\alpha n\})$.*

*Proof.* For any stream $S = (X_1, \ldots, X_n)$, we use the triangle inequality so that

$$|Q_{\mathcal{D}}^q - \tilde{Q}_S^q| \leq |Q_{\mathcal{D}}^q - Q_S^q| + |Q_S^q - \tilde{Q}_S^q| \tag{3}$$

$$\leq \alpha/2 + \alpha/2. \tag{4}$$

$|Q_{\mathcal{D}}^q - Q_S^q| \leq \alpha/2$ follows with probability $1 - \beta/2$ by Corollary B.14 and $|Q_S^q - \tilde{Q}_S^q| \leq \alpha/2$ follows with probability $1 - \beta/2$ by Lemma B.15. The space complexity follows with probability 1 via the deterministic nature of the Greenwald-Khanna sketch. $\square$

**Lemma B.13** (Dvoretzky-Kiefer-Wolfowitz inequality (Dvoretzky et al., 1956)). *For any $n \in \mathbb{Z}_+$, let $X_1, \ldots, X_n$ be i.i.d. random variables with cumulative distribution function $F$ so that $F(x)$ is the probability that a single random variable $X$ is less than $x$ for any $x \in \mathbb{R}$. Let the corresponding empirical distribution function be $F_n(x) = \frac{1}{n}\sum_{i=1}^n \mathbb{1}[X_i \leq x]$ for any $x \in \mathbb{R}$. Then for any $\gamma > 0$,*

$$\mathbb{P}\left(\sup_{x\in\mathbb{R}}|F_n(x) - F(x)| > \gamma\right) \leq 2\exp(-2n\gamma^2).$$

**Corollary B.14.** *For any $q \in (0, 1)$, let $Q_{\mathcal{D}}^q$ be the q-quantile estimate for the distribution $\mathcal{D}$ and $Q_S^q$ be the q-quantile estimate for the sample. Then, $|Q_{\mathcal{D}}^q - Q_S^q| \leq \alpha/2$ with probability $1 - \beta/2$ when $n \geq \frac{2}{\alpha^2} \log 4/\beta$.*

*Proof.* Follows by the DKW inequality (Lemma B.13) where $n \geq \frac{1}{2\gamma^2} \log 4/\beta$ and $\gamma = \alpha/2$. $\qquad\square$

**Lemma B.15.** *For any $q \in (0, 1)$, $\alpha > 0$, $\beta \in (0, 1]$, $\epsilon, \delta \in (0, 1/n)$, there exists an $(\epsilon, \delta)$-differentially private algorithm $\tilde{Q}_S^q$ for computing the q-quantile such that*

$$|Q_S^q - \tilde{Q}_S^q| \leq \alpha/2,$$

*with probability $\geq 1 - \beta$ for stream length*

$$n \geq O\left(\min\{O\left(\frac{R}{\epsilon\sigma\alpha} \log \frac{R}{\sigma\beta}\right), O\left(\frac{R}{\epsilon\sigma\alpha} \log \frac{1}{\beta\delta}\right)\}\right).$$

*Furthermore, with probability 1, $\tilde{Q}_S^q$ uses space of $O(\max\{\frac{R}{\sigma}, \frac{1}{\alpha} \log \alpha n\})$.*

*Proof.* First, by the tail bounds of the Gaussian distribution (Claim B.17), we can obtain that for any $i \in [n]$,

$$\mathbb{P}[|X_i - \mu| > c] \leq 2e^{-c^2/2\sigma^2},$$

so that by the union bound,

$$\mathbb{P}[\exists i, |X_i - \mu| \geq c] \leq 2ne^{-c^2/2\sigma^2},$$

which implies that for any $\beta \in (0, 1]$,

$$\mathbb{P}[\forall i, |X_i - \mu| \leq \sigma\sqrt{2\log 4n/\beta}] \geq 1 - \beta/2,$$

which holds by our sample complexity (stream length) guarantees.

Next, let $r = \lceil R/\sigma \rceil$. [6] Divide $[-R-\sigma/2, R+\sigma/2]$ into $2r+1$ bins of length at most $\sigma$ each. Each bin $B_j$ should equal $((j - 0.5)\sigma, (j + 0.5)\sigma]$ for any $j \in \{-r, \ldots, r\}$. Next run the histogram learner of Lemma B.16 with per-bin accuracy parameter of $\alpha/K$, high-probability parameter of $\beta/2$, privacy parameters $\epsilon, \delta \in (0, 1/n)$, and number of bins $K = 2\lceil R/\sigma \rceil + 1$. We can do this because of our sample complexity (stream length) bounds. Then we obtain noisy estimates $\tilde{p}_{-r}, \ldots, \tilde{p}_r$ with per-bin accuracy of $\alpha/K$. Then any quantile estimate would have accuracy of $\alpha$ (by summing noisy estimates for at most $K$ bins).

Next, we use these bins to construct a sketch (private by DP post-processing) based on the deterministic algorithms of (Greenwald & Khanna, 2004) to, with probability 1, obtain space of $O(\max\{\frac{R}{\sigma}, \frac{1}{\alpha} \log \alpha n\})$. $\quad\square$

**Lemma B.16** (Histogram Learner (Bun et al., 2015; Vadhan, 2017; Karwa & Vadhan, 2018)). *For every $K \in \mathbb{N} \cup \{\infty\}$ and every collection of disjoint bins $B_1, \ldots, B_K$ defined on the domain $\mathcal{X}$. For any $n \in \mathbb{N}$, $\epsilon, \delta \in (0, 1/n), \alpha > 0$, and $\beta \in (0, 1)$, there exists an $(\epsilon, \delta)$-DP algorithm $M : \mathcal{X}^n \to \mathbb{R}^K$ such that for every distribution $\mathbb{D}$ on the domain $\mathcal{X}$, if*

1. *$X_1, \ldots, X_n \sim \mathbb{D}$, $p_k = \mathbb{P}[X_i \in B_k]$ for any $k \in [K]$,*

2. *$(\tilde{p}_1, \ldots, \tilde{p}_K) \leftarrow M(X_1, \ldots, X_n)$,*

3. *$n \geq \max\left\{\min\left\{\frac{8}{\epsilon\alpha} \log \frac{2K}{\beta}, \frac{8}{\epsilon\alpha} \log \frac{4}{\beta\delta}\right\}, \frac{1}{2\alpha^2} \log \frac{4}{\beta}\right\}$,*

*then (over the randomness of the data $X_1, \ldots, X_n$ and of $M$)*

1. *$\mathbb{P}_{X \sim \mathbb{D}, M}[|\tilde{p}_k - p_k| \leq \alpha] \geq 1 - \beta$,*

2. *$\mathbb{P}[\arg\max_k \tilde{p}_k = j] \leq np_j$ if $K \geq 2/\delta$,*

---

[6] Note that this argument is similar to the arguments for Algorithm 1 in (Karwa & Vadhan, 2018).

3. $\mathbb{P}[\arg\max_k \tilde{p}_k = j] \leq np_j + 2\exp(-(\epsilon n/8) \cdot (\max_k p_k))$ *if* $K < 2/\delta$.

**Claim B.17** (Gaussian Tail Bound)**.** *Let $Z$ be a random variable distributed according to a standard normal distribution (with mean 0 and variance 1). For every $t > 0$,*

$$\mathbb{P}[|Z| > t] \leq 2\exp(-t^2/2).$$

**Lemma B.18.** *Algorithm 2 satisfies $(\epsilon, 0)$-DP.*

*Proof.* For any $i \in [n]$, any item $x_i$ can belong in at most one bin. Plus, the global sensitivity of the function that computes the empirical histogram is 2, since changing a single item can change the contents of at most two bins.

As a result, adding noise of $\mathrm{Lap}(0, 2/\epsilon)$ to each bin satisfies $\epsilon$-DP by Theorem B.19. $\qquad\square$

**Theorem B.19** (Laplace Mechanism (Dwork et al., 2006))**.** *Fix $\epsilon > 0$ and any function $f : \mathcal{Y}^n \to \mathbb{R}^K$. The Laplace mechanism outputs*

$$f(y) + (L_1, \ldots, L_K),$$

*$L_1, \ldots, L_K \sim \mathrm{Lap}(0, GS_f/\epsilon)$ where $GS_f$ is the global sensitivity of the function $f$. Furthermore, the mechanism satisfies $(\epsilon, 0)$-DP.*

### B.3 Extension to the Continual Observation setting

In this section we prove Theorem 5.1, which formalizes the privacy and utility guarantees of Algorithm 3 in the continual observation setting. We start by making some definitions that will aid us in our analysis.

**Definition B.20.** We make the following definitions:

1. **Stream Prefix**: Given an input data stream $X = (x_1, \ldots, x_n)$ we define the prefix up to the index $s$ element of this stream $X[1 : s] := (x_1, \ldots, x_s)$. Note that we can overload notation and treat $X[1 : s]$ as a data set by ignoring the order in which the elements arrive.

2. **Checkpoint**: A set $S \neq \emptyset$ is a set of checkpoints for stream $X$ if $\forall j \in S$, $\exists$ value $v_j$ that is a $(\alpha/2)$-approximate $q$-quantile for $X[1 : j]$.

We first observe that if we have an $\alpha/2$ approximate quantile for a given data set, then that estimate remains at least an $\alpha$-approximate quantile for a slightly larger set as well.

**Lemma B.21.** *If $x \in X$ is an $\alpha/2$-approximate $q$-quantile for a data set $X$ (for some $q \in [0, 1]$), then it is an $\alpha$-approximate $q$-quantile for any data set $X' \supset X$ such that $|X'| \leq (1 + \alpha/2)|X|$.*

*Proof.* Since $x$ is an $\alpha/2$-approximate $q$-quantile, we have that

$$(1 - \alpha/2)|X| \leq \underset{X}{\mathrm{rank}}(x) \leq (1 + \alpha/2)|X|.$$

We then have that

$$\underset{X'}{\mathrm{rank}}(x) = \sum_{y \in X'} \mathbb{1}[y \leq x]$$

$$= \sum_{y \in X} \mathbb{1}[y \leq x] + \sum_{y \in X' \setminus X} \mathbb{1}[y \leq x]$$

$$\Rightarrow \sum_{y \in X} \mathbb{1}[y \leq x] \leq \sum_{y \in X'} \mathbb{1}[y \leq x] \leq \sum_{y \in X} \mathbb{1}[y \leq x] + \sum_{y \in X' \setminus X} \mathbb{1}[y \leq x]$$

$$\Rightarrow (1 - \alpha/2)|X| \leq \sum_{y \in X'} \mathbb{1}[y \leq x] \leq (1 + \alpha/2)|X| + (|X'| - |X|)$$

$$\Rightarrow (1 - \alpha/2)|X| \leq \sum_{y \in X'} \mathbb{1}[y \leq x] \leq (1 + \alpha/2)|X| + \alpha|X|/2$$

$$\Rightarrow (1 - \alpha)|X| \leq \sum_{y \in X'} \mathbb{1}[y \leq x] \leq (1 + \alpha)|X|,$$

i.e., $x$ is also an $\alpha$-approximate $q$-quantile for $X'$, as required. $\qquad\square$

We now show that the choice of checkpoints made in the pseudocode of algorithm 3 is valid as per our definition.

**Lemma B.22.** *In Algorithm 3, the set $\{\mathrm{cp}(s') : s' \in [n]\}$ forms a valid set of checkpoints for the data stream $X$. More concretely $v_s$ is an $\alpha/2$-approximate $q$-quantile for $X[1:s]$.*

*Proof.* First we bound the size of the set of checkpoints $\{\mathrm{cp}(s') : s' \in [n]\}$. Since a new checkpoint value is generated only when $s \geq (1+\alpha/2)\mathrm{cp}(s-1)$, it follows that for any new checkpoint where $\mathrm{cp}(s') \neq \mathrm{cp}(s'-1)$, we have $\mathrm{cp}(s') \geq (1 + \alpha/2)\mathrm{cp}(s' - 1)$. Using Theorem 4.1, we set the first checkpoint value to $\frac{24 \log |\mathcal{X}|/\beta^*}{\alpha^* \epsilon^*}$, where $\alpha^* = \alpha/2$, $\beta^*$ and $\epsilon^*$ are the accuracy parameter, the failure probability, and the private parameter that are passed in the calls to `DPExpGK`, respectively. [7] It follows that there are at most $k \leq \log_{1+\alpha/2} n = \frac{3}{\alpha} \log n$ checkpoints using the fact that for all $x > -1$, $\frac{x}{1+x} \leq \log(1 + x) \leq x$.

Since $v_{\mathrm{cp}(s)}$ is the output of `DPExpGK` given a GK sketch with accuracy parameter $\alpha/2$ and privacy parameter $\epsilon^*$ it follows that with probability $1 - \beta^*$ where $\beta^* = \frac{\alpha\beta}{3 \log n}$, $\mathrm{rank}_{X[1:s']} \in (1 - \alpha/2, 1 + \alpha/2)qn$. We now apply the union bound over all private approximate quantile computations at checkpoints and are done. $\quad\square$

We can now prove our main technical result.

**Theorem 5.1.** *Let $\epsilon, \alpha > 0$, $n \in \mathbb{Z}$. For any $\beta \in (0, 1]$, with probability $\geq 1 - \beta$, Algorithm 3 maintains an $\alpha$-approximate $q$-quantile at every point $s$ in the data stream for $s = \Omega\left(\frac{\log n \log |\mathcal{X}|/\beta}{\alpha^2 \epsilon}\right)$. Furthermore, Algorithm 3 satisfies $\epsilon$-DP and has space complexity $\Omega\left(\frac{1}{\alpha^2 \epsilon} \log^2 n \log\left(\frac{|\mathcal{X}| \log n}{\alpha\beta}\right)\right)$.*

*Proof.* To see that Algorithm 3 is $\epsilon$-DP, we observe that the output of this algorithm throughout the data stream can be summarized by its outputs at the checkpoints $\{\mathrm{cp}(s) : s \in [n]\}$ (the points in the stream at which a new checkpoint is reached and a new value released are known publicly, so this suffices for privacy analysis). It follows that there is a choice of $\epsilon^* = \frac{\alpha\epsilon}{3 \log n}$ that gives us an $\epsilon$-DP mechanism.

We now prove the accuracy guarantee. From the second statement of Theorem 4.1 we get that for points in the stream $s$ such that $\mathrm{cp}(s) \geq \frac{48 \log |\mathcal{X}|/\beta}{\alpha \min\{\epsilon^*, 1\}}$, i.e., the first checkpoint, the output $v_s$ will be an $\alpha/2$-accurate quantile for $X[1 : \mathrm{cp}(s)|$. Then, since $|X[1:s]| = s \leq (1 + \alpha/2)\mathrm{cp}(s) \leq (1 + \alpha/2)|X[1:\mathrm{cp}(s)]|$, by Lemma B.21 it follows that $\mathrm{rank}_{X[1:s]}(v_s) \in (1 - \alpha, 1 + \alpha)qn$, i.e., $v_s$ is an $\alpha$-approximate $q$-quantile for $X[1:s]$. $\quad\square$

## C  Additional Experiments

In this section, we include some additional experimental details and results.

**Varying $\epsilon$ for `DPExpGKGumb`:** In Table 1, we vary $\epsilon$ and compare `DPExpFull` to `DPExpGKGumb` in terms of average absolute error and execution time (in seconds). We see that `DPExpGKGumb` is significantly faster than `DPExpFull` in terms of average execution time. However, `DPExpGKGumb` incurs larger error because of the approximation factor of $\alpha = 0.0001$.

---

[7] The last checkpoint might occur at $(1 + \alpha/2)n$. We may ignore checkpoints past $n$.

| $\epsilon$ | DPExpFull (Error) | DPExpGKGumb (Error) | DPExpFull (Time) | DPExpGKGumb (Time) |
|---|---|---|---|---|
| 0.1 | 0.0019796 | 0.002172 | 0.1330166 | 0.028163 |
| 0.5 | 0.0004581 | 0.0013467 | 0.13113595 | 0.02804 |
| 1 | 0.000249894 | 0.00127795 | 0.1335331 | 0.0284041 |
| 5 | 0.00007110 | 0.00151 | 0.1341164231 | 0.028285 |

Table 1: $\alpha = 0.0001$, $n = 10^5$, U(0, 10). Time in Seconds.

**Histogram-Based Algorithm:**  We also implemented the DPHistGK and tested by varying the bin width of the histogram. In general, we observe that DPExpGK has superior performance in terms of minimizing the error for any single quantile query. For example, we obtain averge absolute errors of at least 2.56 for 2 bins and 3.001 for 10 bins for $q = 0.3$ for DPHistGK. This suggests that we can rely on DPExpGK and its variants for general-purpose implementations with minimal error. However, it is possible that certain parameter settings (e.g., bin width) for DPHistGK might yield better performance.

**Continual Observation Algorithm:**  We also implement the continual observation algorithm that builds directly on DPExpGK. We fixed the first stream checkpoint to 10000. For $\alpha = 0.001$ and stream length of $n = 100000$ from $U(0, 10)$, we observe average absolute error of 0.00723 for $\epsilon \in \{0.1, 0.5, 1, 5\}$ over 100 trials. For $\alpha = 0.001$ and stream length of $n = 100000$ from $\mathcal{N}(5, 1)$, we observe average absolute error of 0.00134. However, it is possible that the first few checkpoints might be an important parameter for the error of the sketch in the continual observation setting. We leave the full exploration of this question to future work.

### C.1  Full Space Quantile Computation

Without the bounded space requirement (i.e., space sublinear in the stream length), we can use the exponential mechanism with a score function that uses the entire stream of values $X$. In that case, the sensitivity of the score function is at most 1. We use this as one of the baselines for our experimental validation.

**Lemma C.1.** *Given any insertion only stream*

$$X = (x_1, x_2, \ldots, x_{n-1}, x_n),$$

*the sensitivity of the score function $u$ (under swap differential privacy) is at most 1. i.e., $\delta_u \leq 1$. The function $u$ is defined as $u(X, e) = -|\operatorname{rank}(X, e) - r|$ where $r$ is the approximate $\lfloor q \cdot n \rfloor$ rank of the sketch and $\operatorname{rank}(X, e)$ is the rank of $e$ amongst all values in the stream $X$.*

*Proof.* Let $|X| = n$. The score function becomes $-|\operatorname{rank}(X, e) - n_q|$ where $n_q = \lfloor q \cdot n \rfloor$. Consider two streams with only one element changed: $X, X'$, denoting the element by $x_d$. Then at time $d \leq n$, in the second stream $x'_d$ is inserted instead of $x_d$. In both cases, $n_q$ changes by at most $q$ (in the case of add-remove DP) and for swap DP, $n_q$ remains the same. And for any $e$, $\operatorname{rank}(X', e)$ would differ from $\operatorname{rank}(X, e)$ by at most 1 since the rank of any element can change by at most 1 after adding, deleting, or replacing an item in the stream. Furthermore, for any $n \geq d$, the rank of any $e$ will differ in $X, X'$ by at most 1 replacing $x_d$ with $x'_d$ can displace the rank of any element by at most 1. Also, the term $n_q$ will remain the same. (Note that in the add-remove privacy definition $n_q = \lfloor q \cdot n \rfloor$ would change to either $\lfloor q \cdot (n+1) \rfloor$ or $\lfloor q \cdot (n-1) \rfloor$.)

The "reverse triangle inequality" says that for any real numbers $x$ and $y$, $|x - y| \geq ||x| - |y||$. As a result, $-|\operatorname{rank}(X, e) - n_q| + |\operatorname{rank}(X', e) - n_q| \leq |\operatorname{rank}(X', e) - \operatorname{rank}(X, e)| \leq 1$ for any $e$. $\qquad\square$

