# OpenReview forum: "Bounded Space Differentially Private Quantiles"
_TMLR — Accepted by TMLR_

### Review · Reviewer_B6tU · 2023-02-21

**Summary Of Contributions:**

This paper studies the quantile estimation problem under a combination of small space (streaming) and differential privacy.  The goal is a combination of a summary (which could be maintained in a stream) that is size significantly less than the input size n (in their case roughly log(n)), and has a DP guarantee.
The authors, nor this reviewer, are aware of any other work at this intersection.  Hence, the first algorithm and analysis for this is enough to publication in TMLR.

The algorithm is quite simple:
  * run a standard streaming quantile algorithm (the paper chooses the classic GK sketch)
  * on a query apply the Laplacian mechanism to the answer.
What appears to be crucial (I had to dig into the appendix to figure this out -- I think I am interpreting it correctly) is that the sketch defines a region of values which satisfy the alpha-approximate quantile property.  The exponential DP mechanism then results in a value from a distribution that is:
        (a) uniform over this region
        (b) decays exponentially for values which deviate from this range.


The paper also discussion extensions to:
  * (1) how to apply this mechanism on all possible queries, not just *one* queried point.
  * (2) how to improve bounds when the distribution is well-behaved (is known to be Normal)
  * (3) how to ensure DP while data continues to stream in
  * (4) some empirical studies.
These are all straight-forward.  I have two small concerns:

For (1).  It queries at 1/alpha quantiles adjusting the error guarantees by a factor (1/alpha), and claims these values are enough.  However, I worry what happens if the jth sampled value is larger than the (j+1)th sampled value. Does that cause any issues?  (e.g., what happens if the returned 0.4 quantile is larger than the 0.45 quantile?)

For (4).  The error is measured in the value space, under relative error.  This is what DP cares about.  But streaming quantile summaries typically measure error in the estimates of the CDF.  e.g., if one asks for the 0.35 quantile, and the algorithm returns a value v, then the error is
  |0.35 - CDF(@ v)|
That is, if the CDF at value v is 0.20 (there are only 20% of the value less than v), then the error is 0.15.  The example in Figure 1 shows that small changes in v [e.g., true value = 2.0 (CDF(v) = 0.5), approximate value 1.9999 (CDF(1.9999) = 0.125)].  This is not measured at all.  The way error is presented for DP can seemingly cause a very large discrepancy in this sense.  It may be ok, since it is perhaps the best one can do.  But it would be good to evaluate and explore.
  [Under the distributions the paper considers, I assume it won't be too bad since they are smooth.]


I am also not sure why the paper features the old Greenwald-Khanna sketch.  Sure it is deterministic, but there are much better randomized one.  And since the exponential mechanism (and DP in general) is probabilistic, one can simply set probability of failure in each stage to beta/2, and use Union bound to get a total probability of failure beta.

**Audience:**

Yes

**Claims And Evidence:**

Yes

**Requested Changes:**

 * Theorem 4.1 & Algorithm 1 (the main result) should be:
     - written with much more clear notation, so one could implement and trust the DP proof without digging through the appendix.

 * The paper should be written to be able to use a generic quantile sketch, or explain why that cannot work.  The deterministic argument does not really make sense as pointed out in the last paragraph of Sec 2.1.


**Strengths And Weaknesses:**

## What I liked:

 * The paper seems to introduce a problem at the intersection of two interesting areas.

 * It provides what seems to be a general approach, with proven bounds.



## What I did not like:

 * The notation was very heavy, it seemed unnecessarily so at times.  For instance, d, S, and R are overloaded many times.  And some elements were not always explained in the main 12 pages.  For a couple of examples just in Definition 3.1:
      - what are "neighboring databases"
      - what are $S$ and $\mathcal{R}$?

   and notably
      - $\Delta_u$ in (Algorithm 1 & Thm 4.1) is only explained in the appendix (I think)
      - And in Thm 4.1, what is S in Definition 3.1 in this context?

  * Differential Privacy requires a proof :), but the core proofs are only in the appendix, and not even sketched in the first 12 pages.  I would emphasize putting proofs in the first 12 pages for the main result.  This, IMO, is more important than the algorithm focusing on data from a Normal distribution -- that requires a much stronger assumption on the data ... or Section 7.

  * The paper does not justify the specific focus on the GK sketch when there are many better sketches, and which I think can be treated as just a black box.

---

> ### Author Response · Authors · 2023-03-31
> **Response to Reviewer B6tU**
>
> Thanks to reviewer B6tU for the detailed review and helpful feedback. We will revise our paper, accordingly, to address your concerns. Below we now respond to your comments:
>
> --- *"I am also not sure why the paper features the old Greenwald-Khanna sketch."*
>
> We focus on the Greenwalk-Khanna since it is relatively easy to implement and account for the additional randomness due to DP both in the one-shot and continual observation settings. However, the KLL-sketch (Karmin et al., 2016), as we have noted in the conclusion section, might give better accuracy guarantees in certain regimes. We leave this exploration to future work.
>
> --- *"what are "neighboring databases""*
>
> Thanks for pointing this out. We will clarify this point. Neighboring databases refer to datasets that differ in exactly one row.
>
> --- *"what are S and R?"*
>
> We will clarify that these are sets for the output space of the algorithm.
>
> --- *"$\Delta u$ in (Algorithm 1 & Thm 4.1) is only explained in the appendix (I think)"*
>
> Thanks for pointing this out. We will clarify that it stands for the sensitivity.
>
> --- *"And in Thm 4.1, what is S in Definition 3.1 in this context?"*
>
> $S$ in Definition 3.1 would correspond to the reals ${\mathbb R}$. We can further clarify this point.
>
> --- *"Theorem 4.1 & Algorithm 1 (the main result) should be written with much more clear notation, so one could implement and trust the DP proof without digging through the appendix."*
>
> We can add some more clarifications about the notation and the algorithm.
>
> --- *"The paper should be written to be able to use a generic quantile sketch, or explain why that cannot work. The deterministic argument does not really make sense as pointed out in the last paragraph of Sec 2.1."*
>
> We agree with you about writing the paper to use generic quantile sketches (or to outline such an API). We can add some more clarification about the guarantees needed for a generic quantile sketch (e.g., the GK sketch) to be translated into one that satisfies DP guarantees. Also, we have noted, in the conclusion section and elsewhere, that a randomized sketch might give better guarantees although we might have to account for the additional randomness due to DP.

---

> > ### Comment · Reviewer_B6tU · 2023-04-01
> > **generic quantile sketches**
> >
> > Those proposed changes will help clarify things.
> >
> > I strongly encourage the authors to update the writing to be for a generic quantile sketch -- it will make the results much more lasting and up-to-date.

---

> > > ### Comment · Action_Editors · 2023-04-19
> > > **Generic quantile sketch?**
> > >
> > > Hi authors, one of the main requests from the reviewer was to write things as a generic quantile sketch, keeping GK as an exemplar. It has been noted that this was not done in the latest revision. The reviewer, who is an expert in this area, suggests that it may only take a couple of hours. Do you think this is a reasonable assessment? Do you think it's something that you might be willing to do for this paper?

---

> > > > ### Author Response · Authors · 2023-04-20
> > > > **Definition for generic quantile sketch**
> > > >
> > > > Since the reviewer did not give much detail on what kind of "generic quantile sketch" he/she/they would like, we were unsure of what definition to put forward (i.e., what requirements to satisfy). We have now written a more generic quantile definition (Definition 3.7). The motivation for this definition is as follows:
> > > >
> > > > (1) To allow the sketch summary to answer the single or all-quantiles approximation problem.
> > > >
> > > > (2) Allow for the mergeability property. The GK sketch is not known to be fully mergeable but is **one-way mergeable** (weaker form of mergeability). Note that, in the DP context, mergeability increases sensitivity.
> > > >
> > > > (3) To allow for randomization. e.g., the GK sketch is not randomized but the KLL is both randomized and mergeable (although the variant of KLL that is not mergeable saves an extra log factor in the space complexity).
> > > >
> > > > (4) To allow for other properties we might wish for the sketching algorithm to satisfy (which could be included in the set of functions $\mathcal{P}$).
> > > >
> > > > We have now updated the definition in the paper. We hope this definition suffices.

---

> > > > > ### Comment · Reviewer_B6tU · 2023-04-20
> > > > > **generic quantile sketch**
> > > > >
> > > > > By writing things as a generic quantile sketch, I intended that the paper would:
> > > > >  - identify the minimal properties required by the sketch to make the analysis work, perhaps parameterized by the size, error bounds, etc
> > > > >  - write the algorithm to apply those general properties
> > > > >  - write the final analysis result (i.e., Thm 5.1) in terms of those parameterized properties
> > > > >  - then specify results (e.g., as a corollary) for specific sketches
> > > > > If you want, you could just write them in terms of the KLL sketch since it is optimal.  But the way above will be resilient to all current and future known quantile sketches, one can just check if the sketch can be parameterized in that way.  It seems to be that almost all quantile sketches will satisfy your paradigm.
> > > > >
> > > > > Alternatively, one could specify those requirements, (near Definition 3.7 / Lemma 3.8) and then say that you will use the GK sketch as an exemplar.  Then remind the reader after Theorem 5.1 that a similar (and perhaps improved) bound could be made with another sketch that satisfied the properties.  This would require even less editing of the paper.
> > > > >
> > > > > Thank you for adding Definition 3.7.  However, I find it not specific to what is needed, and have a hard time interpreting what $\phi$ is supposed to mean.  The properties in Lemma 3.8 for the GK sketch are more what I think needs to be generalized.  Can you say that any sketch that achieves such properties could be used in place of GK?  Just about all existing sketches do (with perhaps different bounds for size s) seem to.

---

> > > > > > ### Author Response · Authors · 2023-04-20
> > > > > > **Generalizing the Lemma**
> > > > > >
> > > > > > Thanks for the further clarification. We removed Definition 3.7 and generalized Lemma 3.7. In particular, as you suggested, we stated that "Note that the GK sketch achieves the properties outlined in Lemma 3.7 but any sketch that achieves such properties can be used in place of GK." Also, Lemma 3.7 now has no mention of GK but mentions that there exists a "sketching algorithm" that satisfies those properties. We also removed the size/tuple requirements. Further generalizing beyond this point might lead to downstream issues in the *precise* specification of the utility guarantees of the algorithms---both in the one-shot and continual release settings---that we currently have implemented/evaluated in our paper. We hope Lemma 3.7 can be resilient to current/future quantile sketches.

---

> > > > > > > ### Comment · Reviewer_B6tU · 2023-04-20
> > > > > > > **thanks**
> > > > > > >
> > > > > > > Great.  I think this small change makes the paper much more general.  Thanks for your attention.
> > > > > > >
> > > > > > > I noticed a few spots in the latest revision that it did not properly compile, and there are "??" in place of reference links.  I suggest you recompile once more, and upload again.

---

> > > > > > > > ### Author Response · Authors · 2023-04-20
> > > > > > > > **fixed**
> > > > > > > >
> > > > > > > > thanks -- fixed the broken refs.

---

### Review · Reviewer_89Xd · 2023-03-08

**Summary Of Contributions:**

This paper studies differentially private quantile estimation, with sublinear space complexity. It provides two algorithms in the one-shot setting: one relies on the exponential mechanism, and the other one relies on the histograms. The algorithms both instantiate the Greenwald-Khanna sketch for non-private quantile estimation as the basic building block, which is critical for achieving sublinear space complexity. The paper also provides a theoretical analysis of the space complexity for two algorithms. It then extends the one-shot algorithms to the continual release setting. Some experimental results are given to demonstrate the performance of the algorithms.



**Audience:**

Yes

**Broader Impact Concerns:**

It doesn't require a broader impact statement.

**Claims And Evidence:**

Yes

**Requested Changes:**

I have a few concerns and questions.
1. It's not clear how to choose \Delta_i in DPExpGK. Also, \Delta_u is the sensitivity of the score function. Using the same notation \Delta is confusing.
2. I think the continual release algorithm is somewhat weak since we want \alpha to be very small, then O(\log_{1+\alpha} n/n_min) would be very large. Can you elaborate on the role of n_min, and how you achieve O(\log n/\eps\alpha)?
3. The experiments only show the estimation of the median (q=0.5), I think it's worth adding more results for other choices of q. Moreover, it seems the choice of q does not affect the space complexity, but will it affect the accuracy?
4. I'd suggest adding experiments for the continual release setting.
5. Can DPHistGK extend to answer the empirical CDF?

**Strengths And Weaknesses:**

The paper is well-written. The design of the two algorithms is simple, so practitioners will find it easy to implement. The theoretical analysis is also sound. Overall, I think this is a cute paper.

---

> ### Author Response · Authors · 2023-03-31
> **Response to Reviewer 89Xd**
>
> Thanks to reviewer 89Xd for the detailed review and helpful feedback. We will revise our paper, accordingly, to address your concerns. Below we now respond to your comments:
>
> --- *"It's not clear how to choose \Delta_i in DPExpGK. Also, \Delta_u is the sensitivity of the score function. Using the same notation \Delta is confusing."*
>
> $\Delta_i$ in DPExpGK is chosen non-privately through the Greenwald-Khanna (GK) representation. Please see Appendix A for full details on the GK sketch representation. We will add a note to reference Appendix A. Also, we will disambiguate the use of Delta.
>
> --- *"I think the continual release algorithm is somewhat weak since we want \alpha to be very small, then O(\log_{1+\alpha} n/n_min) would be very large. Can you elaborate on the role of n_min, and how you achieve O(\log n/\eps\alpha)?"*
>
> It is an open question to obtain stronger continual release algorithms for approximate quantiles. The continual release algorithm relies on the observation that for certain $\alpha$ values and dataset sizes, $\alpha/2$ approximate quantiles can be used to obtain $\alpha$ approximate quantiles for any dataset that is $1+\alpha/2$ larger than the original. $n_min$ is the smallest data size used to translate the observation into a DP continual observation algorithm.
>
> --- *"The experiments only show the estimation of the median (q=0.5), I think it's worth adding more results for other choices of q. Moreover, it seems the choice of q does not affect the space complexity, but will it affect the accuracy?"*
>
> In the appendix, we have additional experiments (e.g., for the histogram-based algorithm with $q = 0.3$).
>
> --- *"I'd suggest adding experiments for the continual release setting."*
>
> Thanks for the suggestion. Note that since the continual observation relies on DPExpGK (which we have extensive experiments for), we already have experiments for this setting. We have now added some additional experiments for the continual release setting in the appendix.
>
> --- *"Can DPHistGK extend to answer the empirical CDF?"*
>
> Yes, we believe that DPHistGK can be used to answer the empirical CDF. In particular, once the DP histogram is computed, through preprocessing guarantees of DP, we can compute the empirical CDF. However, the estimates might be quite noisy for a large number of histogram bins with little number of observations. We leave an extensive study of this question to future work.

---

### Review · Reviewer_SR6t · 2023-03-19

**Summary Of Contributions:**

The paper studies the problem of DP quantile estimation under the sublinear space complexity constraint.

To address the research problem, the paper builds 2 lines of algorithms, both of which heavily rely on the Greenwald-Khanna algorithm for producing sketch intervals.

The first algorithm applies the exponential mechanism to privately select the appropriate interval in the GK output. The authors design a scoring function for the intervals to achieve this. The use of the Gumbel-max trick to further improve efficiency is natural.

The second algorithm builds a histogram based on GK intervals, derives an approximate empirical CDF, and releases privatized statistics with the Laplace mechanism.

Authors also extend their algorithms and analysis to the continual release setting and conduct basic numerical experiments with their developed algorithms.

**Audience:**

Yes

**Broader Impact Concerns:**

This work develops and analyzes algorithms for DP quantile estimation. In its current form, the work does not present any long-term ethical concerns.

**Claims And Evidence:**

Yes

**Requested Changes:**

**Technical Soundness:**
I am mainly concerned about the development and analysis of Algorithm 2, as I'm uncertain it's a good/reasonable algorithm for the all-quantiles problem as the authors have suggested. Specific questions
- Under the assumptions in Theorem 4.2, do we get an "all-quantiles" guarantee as defined in 3.6?
- For normal quantile estimation, how does Algorithm 2 DPHistGK compare to the simpler algorithm I describe above?
- Could the authors give some examples to clarify why a general utility guarantee would not hold for DPHistGK?

**Writing clarity:**
- $Q_\mathcal{D}^q$ in def'n 3.5 is undefined.
- Page 6, the authors use the term "confidence interval" to describe the intervals returned from GK, where those intervals aren't really CIs in the frequentist statistics sense.
- the distance function $d$ in eq (2) is undefined.


**Strengths And Weaknesses:**

Strengths:

The paper studies an interesting problem: DP quantile estimation with sublinear space complexity. The authors combine basic sketching algorithms with DP algorithms to solve this problem. Both of these add value to the DP machine learning community. The narrative of the first portion of the paper (up to section 4.1) is clear.

Weaknesses:

**Technical Soundness:**
The algorithm and analysis presented in section 4.2 seem to contradict some of the statements made earlier. E.g.,
- Page 5, before section 4.1, the authors motivate DPHistGK to solve the all-quantiles problem (the guarantee I expect here is something along the lines of $P[ \text{quantile diff} < \alpha, \forall q ] \le \beta$). Yet, even under the normality assumption (given at end of page 7), the theoretical guarantee stated in Theorem 4.2 seems to suggest otherwise ($\forall q, P[ \text{quantile diff} < \alpha ] \le \beta$).
- The theoretical analysis in Section 4.2 doesn't quite type-check with the main setup developed in Section 3.2, since the normality assumption in Section 4.2 would assume that the data universe is not only unbounded but also not finite.

The complexity and utility analysis for Algorithm 2 is useful, but the assumptions are also strong (e.g., normality, known bound R etc). Presumably, under these specific assumptions, a simpler algorithm for estimating the quantile would be to first estimate the mean and variance of the underlying normal and derive the quantile value directly from the converted inverse CDF (there are efficient streaming algorithms for mean and variance estimation). How does this simple algorithm compare to the given DPHistGK in terms of utility (under the same privacy spending)?

---

> ### Author Response · Authors · 2023-03-31
> **Response to Reviewer SR6t**
>
> Thanks to reviewer SR6t for the detailed review and helpful feedback. We will revise our paper, accordingly, to address your concerns. Below we now respond to your comments/questions:
>
> --- *"The theoretical analysis in Section 4.2 doesn't quite type-check with the main setup developed in Section 3.2, since the normality assumption in Section 4.2 would assume that the data universe is not only unbounded but also not finite."*
>
> Thanks for pointing this out. We will update Section 3.2 to reflect that the universe need not be finite.
>
> --- *"The complexity and utility analysis for Algorithm 2 is useful, but the assumptions are also strong (e.g., normality, known bound R etc). Presumably, under these specific assumptions, a simpler algorithm for estimating the quantile would be to first estimate the mean and variance of the underlying normal and derive the quantile value directly from the converted inverse CDF (there are efficient streaming algorithms for mean and variance estimation). How does this simple algorithm compare to the given DPHistGK in terms of utility (under the same privacy spending)?"*
>
> Note that Algorithm 2 makes no assumptions on the data (e.g., normality). Algorithm 2 can handle non-normality. However, the bin sizes of the histogram have to be chosen apriori. We agree that, unlike algorithm 1 (DPExpGK), Algorithm 2 might not be broadly applicable—in terms of utility—to different distributional assumptions. The Theorem, that assumes normality, was a simplified case to show that the histogram-based algorithm might be better in terms of solving the all-quantiles problem. The simpler algorithm you described above also seems to have better performance. However, unlike Algorithm 2, the simpler algorithm (i.e., estimate  mean and variance and use it to calculate quantiles for the normal distribution) you’ve described is specialized to the normal distribution. Another example: unlike the simpler algorithm you’ve described, Algorithm 2 could work on multi-modal distributions. Algorithm 1 is presented as our main general algorithm and we leave the multiple quantiles problem for subsequent work.
>
> --- *"Under the assumptions in Theorem 4.2, do we get an "all-quantiles" guarantee as defined in 3.6?"*
>
> Yes indeed, we can obtain an “all-quantiles” guarantee. Once the DP histogram (which gives a private CDF) is constructed on the data, then obtaining multiple quantiles can be done without any additional privacy cost and is just “post-processing”.
>
> --- *"For normal quantile estimation, how does Algorithm 2 DPHistGK compare to the simpler algorithm I describe above?"*
>
> The simpler algorithm you’ve described above performs better for normal quantile estimation. However, unlike the simpler algorithm you’ve described, DPHistGK can handle non-normal distributions as well. We presented Theorem 4.2 as a simple justification for why one might use Algorithm 2 if the data came from a normal distribution. However, Algorithm 2 does not assume that the data is from any particular distribution.
>
> --- *"Could the authors give some examples to clarify why a general utility guarantee would not hold for DPHistGK?"*
>
> You’re right, a general utility guarantee holds for DPHistGK. However, it is most useful on more concentrated, unimodal distributions that will occupy a few bins of a histogram. Algorithm 2 makes no distributional assumptions on the data. As such a general utility guarantee also holds for the algorithm and carries over from histogram utility guarantees. For example, see histogram learner guarantees (Lemma 2.3 in [1]). Assuming the data is i.i.d. (without any specific assumptions on the distribution type), the histogram learner can estimate each bin to high accuracy assuming a large enough sample size. This guarantee can be used to obtain a private CDF or private all-quantiles estimator. Theorem 4.2 just specializes this technique to the normal distribution—which automatically give us concentration properties on the bins. However, we need not only consider the normal distribution.
>
> --- *"$Q_{\mathcal{D}}^q$  in def'n 3.5 is undefined."*
>
> Thanks for pointing this out. We will update the paper to explicitly define the quantity.
>
> --- *"Page 6, the authors use the term "confidence interval" to describe the intervals returned from GK, where those intervals aren't really CIs in the frequentist statistics sense."*
>
> Indeed, the intervals from GK are not frequentist statistics confidence intervals. We will clarify this point.
>
> --- *"the distance function $d$ in eq (2) is undefined."*
>
> Thanks for pointing this out. We will update the paper to explicitly define the quantity.
>
> [1] Vishesh Karwa, Salil P. Vadhan. Finite Sample Differentially Private Confidence Intervals. ITCS 2018: 44:1-44:9

---

### Comment · Action_Editors · 2023-03-23
**Revisions**

The authors should note that all 3 reviews have been submitted. They are advised to respond to the reviews and revise the paper in line with these comments (potentially with changes in another colour). Starting April 2, the reviewers will be able to submit official recommendations, so doing it before then seems reasonable.

---

### Decision · Action_Editors · 2023-05-11

**Recommendation:** Accept as is

**Comment:**

All reviewers were in agreement that the final paper was satisfactory, and appreciate the revisions made by the authors, particularly with respect to generalizing the result.

**Audience:**

The topic of this paper, differentially private quantile estimation, is clearly of interest to the large differential privacy sub-community within TMLR's audience.

**Claims And Evidence:**

The reviewers initially had some questions and concerns about various technical details in the paper. But the authors clarified and enhanced the paper to address all of these, to the satisfaction of the reviewers.